# Using EUREC[4]A/ATOMIC Field Campaign Data to Improve Trade-Wind Regimes in the Community Atmosphere Model

Skyler Graap[1] and Colin M. Zarzycki[1]

[1]Department of Meteorology and Atmospheric Science, Pennsylvania State University, University Park, PA 16802

**Correspondence:** Colin M. Zarzycki (czarzycki@psu.edu)

**Abstract.** Improving the prediction of clouds in shallow cumulus regimes via turbulence parameterization in the planetary boundary layer (PBL) will likely increase the global skill of global climate models (GCMs) because this cloud regime is common over tropical oceans where low cloud fraction has a large impact on Earth's radiative budget. This study attempts to improve the prediction of PBL structure in tropical trade-wind regimes in the Community Atmosphere Model (CAM) by updat-

ing its formulation of momentum flux in CLUBB (Cloud Layers Unified by Binormals), which currently does not by default allow for upgradient momentum fluxes. Hindcast CAM output from custom CLUBB configurations which permit counter-gradient momentum fluxes are compared to in-situ observations from weather balloons collected during the ElUcidating the RolE of Cloud–Circulation Coupling in ClimAte and Atlantic Tradewind Ocean–Atmosphere Mesoscale Interaction Campaign (EUREC[4]A/ATOMIC) field campaign in the Tropical Atlantic in early 2020. Comparing a version with CAM-CLUBB with a

prognostic treatment of momentum fluxes results in vertical profiles that better match large eddy simulation results. Counter-gradient fluxes are frequently simulated between 950 hPa and 850 hPa over the EUREC[4]A/ATOMIC period in CAM-CLUBB. Further modification to the PBL parameterization by implementing a more generalized calculation of the turbulent length scale reduces model bias and RMSE relative to sounding data when coupled with the prognostic momentum configuration. Benefits are also seen in the diurnal cycle, although more systematic model errors persist. A cursory budget analysis suggests the buoy-

ant production of momentum fluxes, both above and below the jet maximum, significantly contributes to the frequency and depth of countergradient vertical momentum fluxes in the study region. This paper provides evidence that higher-order turbulence parameterizations may offer pathways for improving the simulation of trade-wind regimes in global models, particularly when evaluated in a process study framework.

## 1 Introduction

The increase in atmospheric temperatures caused by anthropogenic greenhouse forcing will inevitably lead to changes in the properties of the land surface and the structures of the atmosphere and ocean. These changes can act to either enhance or diminish the effect of the original forcing and are thus known as positive or negative feedbacks, respectively. Among the feedback mechanisms captured in global climate models (GCMs), those relating to changes in cloud profiles represent the largest source of uncertainty in the simulated climate response to increased greenhouse gas concentrations (Ceppi et al., 2017).

Low clouds reflect a significant portion of incoming shortwave radiation but emit longwave radiation at a rate comparable to the surface given the similarity in temperature. This leads to what is called the 'low cloud radiative feedback' whereby an increase in low cloud cover has a net cooling effect on the surface by preventing solar warming, while still allowing for radiational cooling. Near-surface cumulus and stratocumulus clouds are among the most important clouds for this feedback given that they have a sufficient optical depth to prevent sunlight from reaching the surface, can exist at low latitudes that experience high insolation, and can cover large surface areas. Changes in low cloud fractions in the tropics have been described by Ceppi et al. (2017) as one of the three main components of the global cloud feedback in GCMs.

The global scale atmospheric circulation that eventually gives rise to low clouds in the tropics is the Hadley circulation which features rising motion near the equator and sinking motion in the subtropics. This leads to easterly winds (known as the trade winds) at the surface and westerly winds aloft in the tropics. Within this cell regions exist where different large-scale patterns of clouds, known as cloud regimes, tend to arise repeatedly. One of these is the tropical trade-wind cumulus regime, characterized by the formation of many small separate cumulus clouds as a result of shallow convection in the boundary layer over tropical oceans (Ruppert, 2016). Poleward of this cloud regime, in a region known as the subtropical stratocumulus to trade cumulus transition (STCT), there is a gradual transition as the shallow cumulus clouds feed into an overlying stratocumulus layer (Stevens et al., 2002). Poleward of this, the stratocumulus layer breaks up. A large portion of stratocumulus clouds found over subtropical oceans are associated with the transitional regime and thus the STCT has a large impact on the overall climate system cloud-radiative feedback (Stevens et al., 2002; Trenberth et al., 2001). Improvements in GCM prediction of boundary layer structure in the tropical trade-wind regime could improve not only the representation of cloud cover changes locally, but also the prediction of downstream cloud cover change in the STCT where the shallow cumulus clouds feed into a broader stratocumulus layer.

The structure of the PBL is determined in large part by turbulent vertical fluxes which work to redistribute quantities like heat, moisture, and momentum. This turbulence occurs at scales much smaller than typical grid spacing of GCMs and must be parameterized. The vertical flux of horizontal momentum (henceforth simply "vertical momentum flux") can be thought of as the horizontally averaged covariance between the horizontal wind and the vertical wind ($\overline{u_h'w'}$) where $u_h$ is either the zonal ($u$) or meridional ($v$) component of the wind. In most GCMs, the time tendency of $\overline{u_h'w'}$ is parameterized with diagnostic eddy diffusivity (commonly referred to as "K Theory" (Berkowicz and Prahm, 1979; Stensrud, 2007)). This turbulence closure defines $\overline{u_h'w'}$ as the product of the existing vertical gradient in horizontal momentum and a coefficient denoted as $K$. Such a closure can only act to move existing horizontal momentum to an altitude with less momentum (downgradient flux). Recently, it has been shown in large eddy simulations (LES) that momentum fluxes moving in the opposite direction – upgradient fluxes working to move momentum to altitudes with greater horizontal momentum, also referred to as countergradient fluxes – can occur in tropical shallow convection (Larson et al., 2019; Dixit et al., 2020; Helfer et al., 2021). In order for GCMs to capture these upgradient fluxes, they must prognose $\overline{u_h'w'}$. Such a parameterization includes many different source and sink terms in its calculation of $\overline{u_h'w'}$ time tendency, with each term being related to a physical process.

Larson et al. (2019) (henceforth L19) attempted to model $\overline{u_h'w'}$ in marine shallow cumulus layers in a single-column model using data from several field campaigns (including the Barbados Oceanographic and Meteorological Experiment (BOMEX),

which took place over the tropical North Atlantic (Holland and Rasmusson, 1973)). Their model utilizes the higher-order Cloud Layer Unified by Binormals (CLUBB) parameterization and is run in both a mode that only allows downgradient diffusion and a mode that prognoses $\overline{u_h'w'}$. They found that the prognostic momentum configuration was better able to recreate the structure of wind profiles described by an LES run based on the field campaign. LES simulations are integrated at a much higher spatial resolution than operational models and can serve as a spatiotemporally-continuous 'bridge' to point observations that are limited in space and time. The vertical profile of momentum during BOMEX featured a characteristic easterly jet near the top of the boundary layer and the prognostic momentum run was able to recreate the 3-layer structure of $\overline{u_h'w'}$ described by the LES where there is downgradient $\overline{u_h'w'}$ from the surface to near the jet maximum, upgradient flux in the few hundred meters above this jet maximum, and weak $\overline{u_h'w'}$ above this layer.

Similarly, Dixit et al. (2020) (henceforth D20) found upgradient $\overline{u_h'w'}$ in the cloud layer of a tropical shallow convection regime in their investigation of vertical momentum transport using multi-day large eddy models with data from the BOMEX and RICO (Rain in Shallow Cumulus Over the Ocean (Rauber et al., 2007)) field campaigns, both of which took place in the western tropical North Atlantic. Their analysis reveals that these upgradient fluxes are driven by non-hydrostatic pressure gradients and horizontal circulations generated by convection. The effects of these mesoscale dynamics can therefore not be represented by downgradient diffusion alone.

Helfer et al. (2021) also noted upgradient momentum fluxes in their LES simulations run for the tropical North Atlantic in a time period corresponding to the NARVAL (Next-generation Aircraft Remote-sensing for VALidation studies) flight campaign in December 2013 (Vial et al., 2019). They demonstrated that these upgradient fluxes could not be captured by pure $K$ theory based on their calculated profiles of what the coefficient $K$ would have to be as derived by dividing $\overline{u_h'w'}$ by the existing vertical gradient in horizontal momentum ($\frac{dU}{dz}$), sometimes referred to as 'effective diffusivity' (Bryan et al., 2017; Nardi et al., 2022). These profiles showed that negative $K$ would be required (i.e. upgradient fluxes are occurring) for both $u$ and $v$, in certain layers of a vertical structure similar to that found in L19, particularly in the winter. These profiles were calculated for the innermost grid of their LES hindcasts which consisted of multiple nested domains and were ultimately forced by reanalysis data.

This study seeks to build on the findings of L19 by using data from a more recent and intensive process study (Cronin et al., 2009) that took place in generally the same region as BOMEX and RICO (the joint ElUcidating the RolE of Cloud–Circulation Coupling in ClimAte and Atlantic Tradewind Ocean–Atmosphere Mesoscale Interaction Campaign (EUREC[4]A/ATOMIC) field campaign) to evaluate how prognosing, rather than diagnosing, $\overline{u_h'w'}$ affects a three-dimensional GCM's performance in predicting boundary layer structure in tropical trade-wind regimes. Here we focus on the Community Atmosphere Model (CAM), a component of the Community Earth System Model (CESM). Several experimental versions of CAM are created, each of which implements CLUBB and includes a prognostic eddy diffusivity that uses a Reynolds averaging closure. The difference between separate prognostic momentum runs lies in how the vertical turbulent length scale is estimated. Output from these versions of CAM, as well as from the default unmodified version, are compared to state variable data from 1,546 weather balloon soundings collected during the six-week EUREC[4]A/ATOMIC field campaign.

## 2 Data and Methods

All of the observational data used in this study to evaluate model predictions come from the EUREC[4]A/ATOMIC mass data collection field campaign. EUREC[4]A/ATOMIC was conducted over the tropical North Atlantic Ocean just east of Barbados in January and February 2020 (Stevens et al., 2021). Boundary layer measurements collected for this field campaign are of higher resolution and quality than previous field campaigns in the same region like BOMEX and RICO (Savazzi et al., 2022). While recent, these data are beginning to be exploited to evaluate model performance in this region. For example, Savazzi et al. (2022) used the weather balloon sounding, dropsonde, and lidar data from EUREC[4]A/ATOMIC to characterize the wind profile structure of the boundary layer and to evaluate the performance of the Integrate Forecast System (IFS) of the European Centre for Medium-Range Weather Forecasts (ECMWF) along with the related ERA5 reanalysis data in the prediction of boundary layer wind profiles during EUREC[4]A/ATOMIC. Some of the techniques employed by Savazzi et al. (2022) to evaluate the performance of IFS using this data set are used here to evaluate the performance of CAM.

### 2.1 EUREC[4]A/ATOMIC Sounding Data

During the EUREC[4]A/ATOMIC campaign, radiosondes attached to weather balloons were launched from four ships and the Barbados Cloud Observatory (BCO) over the course of 43 consecutive days from 8 January to 19 February 2020. For most of this period, soundings were attempted every four hours from all five stations, but not all stations reported every day (see Figure 1 in Stephan et al. (2020) for a complete time series of all balloon launches). Most balloon launches recorded data during both the ascent of the balloon and the descent of the radiosonde with a parachute after the balloon burst, however only data from the ascents are used here because the descent data are likely less reliable given the rapid fall speed. The four ships were moving during the field campaign, but at all times, all ships were located somewhere between 6 and 16 ° N and between 50 and 60 ° W (see Fig. 2 in Stephan et al. (2020) for a complete time series of ship locations). All stations launched Vaisala RS41-SGP radiosondes and recorded horizontal wind ($u$ and $v$ components), temperature ($T$), relative humidity, and pressure at even intervals of 10 meters altitude starting at 30 or 40 meters above the surface until balloon burst, up to a maximum altitude of 31 km. Additionally, 47 radiosondes of Meteomodem type M10 were launched from one of the ships (the L'Atalante) without parachutes (Stephan et al., 2020). These soundings also reported data every 10 meters.

### 2.2 CAM Configurations

The version of CAM studied here is CAM version 6 (Bogenschutz et al., 2018; Gettelman et al., 2019). This corresponds to the configuration of CAM in the CESM version 2 release (Danabasoglu et al., 2020) that was used to generate the simulation submitted to the Coupled Model Intercomparison Project version 6 (CMIP6), with two differences. First, we use the spectral element (SE) dynamical core (Lauritzen et al., 2018) on an unstructured cubed-sphere grid with nominal 1° (111km, also referred to as CAM-SE's ne30np4 grid) horizontal grid spacing. This is in lieu of the CAM6 default finite-volume dynamical core. Second, we use 58 vertical levels with finer grid spacing in the atmospheric boundary layer compared to CAM6's default 32 layers. The height of the lowest model level is approximately 22m and the model top is approximately 40 km. The most

significant parameterization change in CAM6 from predecessor versions is the addition of CLUBB as a unified turbulence scheme to replace otherwise separate boundary layer, shallow-convection, and macrophysics parameterizations (Bogenschutz et al., 2013; Danabasoglu et al., 2020). CLUBB is a high-order closure that represents moist turbulence with a simple multivariate probability density function to describe sub-grid variations in potential temperature ($\theta$), water vapor mixing ratio ($Q$), and vertical velocity ($w$) (Golaz et al., 2002; Larson, 2022). CLUBB is discretized in the vertical by centered differencing or else upwind differencing on a staggered grid and implements a semi-implicit time stepper where the time stepping method is simple backward Euler (Larson, 2022). State variables solved for in the dynamical core of CAM include air temperature ($T$), $Q$, $u$, $v$, and surface pressure ($p_s$). Since CAM is a hydrostatic model, the vertical pressure velocity ($\omega$) is diagnosed from the continuity equation. Other quantities, such as turbulence outputs, are solved for in the model's subgrid parameterization suite.

In this study, CAM is initialized twice daily (00Z and 12Z) with the 0.25° ERA5 (Hersbach et al., 2020) reanalysis data using the Betacast software package, first described in Zarzycki and Jablonowski (2015). To initialize the model, the ERA5 state field is mapped to the CAM grid using high-order remap operators, with the hydrostatic correction of Trenberth et al. (1993) applied to balance the model state against CAM's lower-resolution orography. The model was run with prescribed ocean and ice fields using observations from NOAA's Optimum Interpolation (OI) dataset (Reynolds et al., 2002) and are fixed for the duration of the hindcasts. The model's land state was generated by using three-hourly surface forcing derived from ERA5 to drive an offline version of the Community Land Model (CLM) for the 12 months before the EUREC[4]A/ATOMIC period. Subsequent land initializations leverage the 12-hour land surface forecast from the previous cycle as in Zarzycki and Jablonowski (2015). This creates a surface state consistent with atmospheric observations during the period prior to the simulation, although it is worth noting that we anticipate impacts from the land surface model are negligible given the domain of interest and duration of the hindcasts. The model is then integrated for 72 hours in different configurations providing output every 30 minutes for each day of the EUREC[4]A/ATOMIC Core Period (8 January - 19 February 2020). In order that CAM output from runs initialized 0, 1, and 2 days prior are available for all days during the field campaign in addition to approximately a week following it, CAM is initialized for the three days leading up to the campaign and then every day during it (from 00Z 5 January 2020 to 12Z 25 February 2020), resulting in 104 initializations for each configuration discussed below. All simulations were completed using the Cheyenne supercomputer, maintained at Computational Information Systems Lab and funded by National Science Foundation (CISL, 2019).

### 2.2.1 Diagnostic Versus Prognostic Configurations

The unaltered version of CAM described above (henceforth known as "eddy-diffusivity, original length scale" (ED-O) or "the default run") is the run against which the other configurations of CAM are compared. In this configuration, $\overline{u_h' w'}$ are calculated using a diagnostic eddy diffusivity approximation:

$$\overline{u' w'} = -K_m \frac{\partial \overline{u}}{\partial z} \tag{1}$$

$$\overline{v'w'} = -K_m \frac{\partial \overline{v}}{\partial z} \tag{2}$$

where $K_m$ is a tunable transfer coefficient (Golaz et al., 2002). Here, $\overline{u_h'w'}$ is simply a function of the vertical shear of the resolved horizontal wind. The turbulent transfer coefficient is defined to be positive, and thus, such a diagnosis is incapable of producing $\overline{u_h'w'}$ that acts to move momentum 'up' the existing gradient.

An experimental CAM configuration is created by replacing the eddy diffusivity closure by using a higher order closure described by Eq. 3 to prognose $\overline{u_h'w'}$. This closure, which calculates the time tendency of $\overline{u_h'w'}$ by considering several source and sink terms, can be considered an incomplete third order closure since $\overline{w'^3}$ is prognosed by CLUBB (Larson, 2022; Larson et al., 2019; Nardi et al., 2022). We refer to this as the "prognostic momentum, original length scale" (PM-O) configuration. We stress that, aside from this change, all other components of ED-O and PM-O are identical. Unless otherwise specified, all model settings and configurations are the default used in CAM6 for the CESM2 release.

$$\frac{\partial \overline{u_h'w'}}{\partial t} = \underbrace{-\overline{w}\frac{\partial \overline{u_h'w'}}{\partial z}}_{1} - \underbrace{\frac{1}{\rho}\frac{\partial \rho \overline{w'^2 u_h'}}{\partial z}}_{2} - \underbrace{(1-C_{uu,shear})\overline{w'^2}\frac{\partial \overline{u_h}}{\partial z}}_{3} - \underbrace{(1-C_7)\overline{u_h'w'}\frac{\partial \overline{w}}{\partial z}}_{4} + \underbrace{(1-C_7)\frac{g}{\theta_{vs}}\overline{u_h'\theta_v'}}_{5} - \underbrace{\frac{C_6}{\tau}\overline{u_h'w'}}_{6} - \underbrace{\epsilon_{u_hw}}_{7} \tag{3}$$

Here, $\rho$ is the air density, $g$ is gravity, $\theta_v$ is virtual potential temperature, $\tau$ is the eddy turnover time scale, and $C_{uu,shear}$ is an empirical constant with a default value of 0.3. $C_6$ and $C_7$ are also tunable values that are left unchanged for ED-O and PM-O from CAM6 defaults. $C_7$ is set to 0.5 and $C_6$ is a skewness function described in Eq. 5 of (Guo et al., 2014), where $C_{6rt}$=$C_{6thl}$ is 6, $C_{6rtb}$=$C_{6thlb}$ is 4, and $C_{6rtc}$=$C_{6thlc}$ is 1. The terms here describe how $\overline{u_h'w'}$ can either be generated or dissipated through 1) advection by the mean vertical wind, 2) turbulent advection by perturbations in the vertical wind, 3) turbulent production by updrafts and downdrafts, 4) turbulent production from pre-existing $\overline{u_h'w'}$ existing in a vertical gradient in the mean vertical wind, 5) buoyant production, 6) a 'return-to-isotropy' adjustment that has the magnitude of $\overline{u_h'w'}$ decay over time, and 7) a residual dissipation term (Nardi et al., 2022). The derivation of this equation is described in Appendix A alongside additional turbulence closures for the remaining unsolved terms.

In PM-O, the eddy turnover time scale, $\tau$, which describes the rate of decay in the 'return-to-isotropy' term, is calculated as the vertical turbulent length scale ($L$) divided by the square root of turbulent kinetic energy (TKE or $\overline{e}$) as defined in Eq. 25 of Golaz et al. (2002):

$$\tau = \frac{L}{\sqrt{\overline{e}}} \tag{4}$$

and TKE is calculated from variances of each wind component (each predicted by CLUBB):

$$\overline{e} = \frac{1}{2}(\overline{u'^2} + \overline{v'^2} + \overline{w'^2}) \tag{5}$$

This turbulent length scale is described by the mean of the upward and downward distances a parcel could travel before its change in potential energy from buoyancy equals the total turbulent kinetic energy that it started with (Golaz et al. (2002) Eqs. 36, 37, 38). This formulation of $\tau$ depends only on turbulent kinetic energy (TKE) and atmospheric stability. In PM-O, $L$ is calculated as described above and $\tau$ is diagnosed from that value of $L$ and TKE. The same is the case in ED-O.

## 2.3 Prognostic Configurations with Experimental Vertical Turbulent Length Scale Estimates

To explore the impact of the shape of the turbulence profile (i.e., the shape of either $L$ or $\tau$ profiles), we explore an alternative treatment of $\tau$ described in Guo et al. (2021). Here, $\tau$ can be calculated using a set of 'building blocks' describing the dissipation of turbulent eddies:

$$\frac{1}{\tau} = \underbrace{C_{\tau,bkgnd} \frac{1}{\alpha}}_{1} + \underbrace{C_{\tau,sfc} \frac{u^*}{\kappa} \frac{1}{(z - z_{sfc} + d)}}_{2} + \underbrace{C_{\tau,shear} \sqrt{\left(\frac{\partial \overline{u}}{\partial z}\right)^2 + \left(\frac{\partial \overline{v}}{\partial z}\right)^2}}_{3} + \underbrace{C_{\tau,N2} \sqrt{N^2}}_{4} \tag{6}$$

In this equation (the sum of Eqs. 19 and 20 in Guo et al. (2021)), $\alpha$ is a constant (1000 s), $u^*$ is the friction velocity, $\kappa$ is the Von Karman constant, $N$ is the Brunt-Väisälä frequency, $d$ is a small displacement height, and $C_{\tau,bkgnd}$, $C_{\tau,sfc}$, $C_{\tau,shear}$, and $C_{\tau,N2}$ are all empirical constants. This equation considers 1) a background dissipation rate, 2) dissipation due to frictional effects near the surface, 3) dissipation due to vertical wind shear, and 4) dissipation in a stable atmosphere (set to 0 in buoyantly unstable and neutral layers). Each term here includes a different tunable coefficient (i.e., the $C_\tau$ terms).

We determine tuning coefficients for this configuration using a Nelder-Mead optimization (Nelder and Mead, 1965). Specifically, a set of very short (48-hour) hindcasts initialized on January 1st, 2012 is run, optimizing various tunable parameters in CLUBB to minimize the difference in the predicted wind field after 2 days when compared against ERA5 reanalysis at the same time. Optimization is completed relative to global ERA5 reanalysis data rather than the local EUREC$^4$A/ATOMIC data to ensure a reasonable global simulation. We set $C_{\tau,bkgrnd}$ to 0.45, $C_{\tau,sfc}$ to 0.04, $C_{\tau,shear}$ to 0.20, $C_{\tau,N2}$ to 0.10, and $C_{uu,shear}$ to 0.005. We also set $C_{uu,buoy}$ to 0.30, $C_{\tau,N2,clr}$ to 0.90, $C_{\tau,N2,wp2}$ to 0.20, and $C_{\tau,N2,xp2}$ to 0.15. The last four parameters are not included in the equations mentioned thus far, but $C_{uu,buoy}$ serves as a parameter in the CLUBB equation for $\overline{w'^2}$ and $C_{\tau,N2,clr}$, $C_{\tau,N2,wp2}$, and $C_{\tau,N2,xp2}$ all serve as subtle tunings on $C_{\tau,N2}$. $C_6$ is reduced to 2 and treated as a constant to better recover the tunings in Guo et al. (2021). We emphasize that with this configuration it is only a scaling factor and not treated as a tunable parameter (Vince Larson, personal communication, December 2021). We also note that this simple optimization process is not meant to replace more formal model tuning (Hourdin et al., 2017), but rather, to provide a plausible configuration with respect to simulated wind profiles for this study.

The relationship between $L$ and $\tau$ described in Eq. 4 is applied, although $L$ is now diagnosed from turbulent kinetic energy and $\tau$ as:

$$L = \tau * \sqrt{\overline{e}} \tag{7}$$

**Table 1.** Description of four CAM configurations described in this paper. The first column represents the abbreviated experiment ID that is used in the figures and text. The momentum flux treatment indicates whether the default 'eddy diffusivity' is used for momentum fluxes or whether the 'prognostic momentum' treatment in Eq. 3 is applied. The length scale treatment indicates whether the turbulent length scale is calculated using the 'original' formulation described in Golaz et al. (2002) or as diagnosed using the 'experimental' method following Guo et al. (2021).

| Exp. ID | Momentum flux treatment | Length scale treatment |
|---------|------------------------|------------------------|
| ED-O | Eddy diffusivity | Original |
| PM-O | Prognostic momentum | Original |
| ED-X | Eddy diffusivity | Experimental |
| PM-X | Prognostic momentum | Experimental |

That is, $\tau$ is computed first and $L$ is diagnosed using this in combination with TKE (Larson, 2022). Henceforth, configurations that use Eq. 6 to calculate $\tau$ (and thus $L$) will be referred to as the 'experimental length scale' runs and are denoted by the letter 'X.' We assess this with both the eddy diffusivity and prognostic momentum formulations from above, resulting in ED-X and PM-X, respectively. We note that $\tau$ does appear in other prognostic CLUBB equations (e.g., turbulent fluxes of scalars) and therefore impacts additional prognostic quantities in the PBL beyond just $\overline{u'_h w'}$ (Larson, 2022). The four configurations explored here are described in Table 1.

### 2.4 Comparison to Observational Soundings

#### 2.4.1 Interpolation of CAM Output

In order to directly compare model output to observational data, model estimates of state variables are calculated for every point reported for every sounding. This is done for every model configuration where 1-day lead time predictions are used (24-48 hours after model initialization) to reduce forecast error and better constrain the simulations based on the initial conditions. Similar results are found when 2-day leads are considered instead (not shown). The profiles are found by taking data from only the model column nearest a sounding and linearly interpolating the vertical profiles of $T$, $Q$, $u$, and $v$. The 'nearest' model column is calculated as that with the smallest great circle distance from the latitude and longitude reported by a balloon at 1 km geopotential height, or if no data were reported for this level, the next lowest altitude for which coordinates are reported. 1 km geopotential height is chosen as the reference point for each sounding because this study mainly focuses on the lowest 2.5 km of the atmosphere. Soundings that do not report any data for altitudes above 1 km are not considered in this study.

Each sounding profile is compared to a purely vertical profile in the model output, but this is reasonable since ascent rates were rapid enough and horizontal wind speeds were slow enough that balloons tended to drift only around 10 km horizontally in the lowest 5 km altitude (the layer of focus), while the nearest model columns are separated by approximately 100 km. Similarly, each observational profile is compared only to model output from the single timestep that is nearest in time to when the sounding reached 1 km geopotential height. This is reasonable since typical balloon ascent rates were 3 to 5 m/s (or

about 1 km in 3 to 5 minutes) and model timesteps are 30 minutes apart. Once a model timestep and column are chosen for a particular sounding, the interpolated vertical profile used in the direct comparison is generated. Since CAM6 uses a hybrid sigma-pressure vertical coordinate, the heights at which CAM data are output can vary between columns and time steps. These reporting altitudes are found for each column and timestep that were chosen to correspond to an observational sounding in each model run. The vertical grid spacing of CAM is approximately 50 meters near the surface, 250 meters at 2 km altitude, and 500 meters at 5 km altitude. This is much coarser than observations, which report every 10 meters. State variables from model output are interpolated to each of these 10-meter levels by taking the linear vertical distance-weighted average of those values reported at the nearest two model levels. Those observational points that lie below the lowest model level simply take the value of that lowest level. There is no analog to this at high altitudes since model output is reported for higher altitudes than all soundings. For each interpolated model prediction that corresponds to a point in the observations, a bias is calculated for each state variable predicted. This is done by simply subtracting the value measured by the observation from that value predicted by the model.

### 2.4.2 Statistical Profile Calculations

Mean, median, and 25th/75th percentile profiles for state variables in observations are all estimated for the whole domain space and time by calculating those metrics at each 10-meter altitude level over all soundings during the campaign. These statistical profiles are also created for the output of each model configuration and lead time by performing the calculations on the corresponding state variable model output that has been interpolated to the observations' 10-meter grid spacing.

These profiles are calculated both for all times of day (by including all soundings) and for particular times of day, by only including soundings whose launch times fit within particular hours of the day. Specifically, eight sets of time-of-day-specific profiles are created, each of which only takes into account those data that were collected by balloons launched during particular non-overlapping 3-hour increments, beginning with 00:00-03:00 UTC (02:00-05:00 local time).

Mean profiles are also created that estimate the vertical profiles in model bias and root mean squared error (RMSE). Bias profiles are simply created by averaging the aforementioned biases calculated at each point, while RMSE profiles are created by, for each altitude, taking the square root of the sum of the squares of each bias from every sounding at that altitude.

### 2.5 Large Eddy Simulations

To provide a bridge between the observed profiles and the highly parameterized CAM simulations, we also generate a model reference simulated with a large-eddy configuration of the Cloud Model 1 (CM1) (Bryan and Fritsch, 2002; Bryan and Rotunno, 2009). While the standard BOMEX LES test case (for example, that run in L19) generates domain-averaged profiles that are qualitatively similar to those observed during EUREC[4]A/ATOMIC, the atmosphere was slightly drier, slightly cooler, and had stronger $u$ and $v$ wind components during the field study of interest here.

To create a more consistent proxy, we begin with the BOMEX test case as described in Siebesma et al. (2003). The horizontal and vertical grid spacings of CM1 are 100 m and 50 m, respectively. The domain extent is 6.4 km x 6.4 km in the horizontal and 3 km in the vertical and we update the Coriolis parameter to be $f = 0.353$ x $10^{-4}$ s$^{-1}$ to represent the study region. Instead of

analytic, idealized profiles, we initialize the model with the mean $u$, $v$, $T$, and $Q$ soundings observed during the field campaign. We prescribe an initial surface pressure of 1015.6 hPa, a surface potential temperature of 298.155 K, and a surface water vapor mixing ratio of 15.9 g/kg. We then use ERA5 to estimate large-scale forcing during the campaign period. We specify a vertical velocity ($w$) profile that linearly decreases from 0 at the surface to -0.25 cm/s at 800 m. The profile is constant at -0.25 cm/s from 800 m to 1800 m, and it decreases linearly from -0.25 cm/s at 1800 m to -0.6 cm/s at 3000 m. To mimic a large-scale pressure gradient, we apply a background geostrophic wind. The zonal component $u_g$ increases linearly from -10.5 m/s at the surface to -2.5 m/s at 3000 m. The meridional component $v_g$ decreases linearly from -1 m/s at the surface to 1 m/s at 1500 m and remains at 1 m/s above that. All remaining configuration options – including specified radiative cooling and low-level drying tendencies – are kept the same as in Siebesma et al. (2003).

We average the simulated output between hours 2 and 6 over the LES domain, similar to what is commonly done for BOMEX evaluations. These profiles are referred to as 'CM1' for the remainder of this paper. We emphasize that we only use this simulation to contextualize the comparison of the CAM results described here with observations taken during EUREC[4]A/ATOMIC. While this CM1 configuration produces a simulation that is well-matched to observed soundings, we acknowledge further improvement or refinement of the model setup may be possible. More detailed budget analyses to better understand the turbulent evolution of quantities in the boundary layer are a target for future research. We also refer interested readers to Narenpitak et al. (2021), Dauhut et al. (2023), and Schulz and Stevens (2023), all of which performed LES simulations using a variety of configurations to investigate the distributions and organization of shallow convective clouds during the EUREC[4]A/ATOMIC study period.

## 3 Results of the Addition of Prognostic Momentum

### 3.1 Momentum Profiles

We first investigate the impact on simulated profiles by replacing parameterization of $\overline{u_h'w'}$ by eddy diffusivity with the prognostic equation (Eq. 3). It can be seen in Fig. 1a that the default version of CAM (ED-O, red dotted line) tends to overestimate the magnitude of the easterly winds at most altitudes below 2.5 km and places the easterly jet maximum at a higher altitude when compared to EUREC[4]A/ATOMIC observations (solid black line) and the CM1 LES results (solid gray line). Given the limited number of soundings that report below 40 m, mean profiles observational profiles below 40 m are not representative of the domain and have been removed from all plots in this study. The same goes for the corresponding plots of errors calculated from those observations.

When adding the prognostic $\overline{u_h'w'}$ formulation in configuration PM-O (dashed green line), the jet maximum becomes stronger in magnitude by around 0.5 m/s but narrower in depth meaning that the vertical gradient of $u$ becomes steeper in the region of the jet. Above this layer, the strong easterly wind bias in ED-O is reduced in PM-O. Although the easterly bias is increased by up to 0.25 m/s in PM-O at altitudes below the jet maximum, the maximum bias in $u$ is actually around 0.2 m/s smaller in PM-O than ED-O, and RMSEs are reduced by up to 0.3 m/s at altitudes between 1 and 2 km in PM-O (see Fig. 2). Biases and RMSEs can become large below 200 m because very few model levels are present in this layer where real-world

conditions can vary significantly with height. Model predictions at these altitudes are highly sensitive to the surface layer for-
mulation, which is not the focus of this study. We also note that winds are generally too strong throughout the lowest 2.5km.
Ignoring the Coriolis force, a turbulence parameterization only rearranges the wind profile in the vertical. This may also imply
that the surface is not inducing enough drag on the lowest model level, although we leave this evaluation for future work.

Figure 1b shows the profiles of $\overline{u'w'}$. No observational profiles exist for turbulence covariances since only state variables are
measured by the radiosondes in EUREC[4]A/ATOMIC. While some aircraft observations of such fluxes were collected as part
of the field campaign (Brilouet et al., 2021), these flights covered a small time window of the campaign and observations were
generally taken along horizontal surfaces. However, the turbulent fluxes as simulated by CM1 are shown in gray for reference.
Below the jet maximum, both ED-O and PM-O show similar $\overline{u'_h w'}$. $\overline{u'_h w'}$ differ greatly above the altitude of the jet maximum
(above approximately 800 m). Both profiles feature negative $\overline{u'w'}$ at these altitudes, but the magnitude 'overshoot' (i.e., the
magnitude of negative $\overline{u'w'}$ values before returning towards 0 with height) is much greater for ED-O.

These $\overline{u'w'}$ profiles are qualitatively very similar to analogous results described in Fig. 8 of L19. They compared results
from a prognostic $\overline{u'_h w'}$ idealized single-column model and an LES running the BOMEX test case. The implementation of
prognostic $\overline{u'_h w'}$ made the easterly jet more narrow and reduced the magnitude of negative $\overline{u'w'}$ above the jet maximum, which
resulted in better agreement with their LES runs (similar to our finding of a better match to CM1 here), which is assumed to
be a physically-based reference. This, along with observations of $u$ and $v$ in our study being structurally similar to the LES-
derived profiles in L19, suggests the addition of prognostic $\overline{u'_h w'}$ improves the realism of how the jet is simulated in PM-O.
The behaviors seen in highly constrained single-column simulations and idealized LES runs can be reproduced in short-term
initialized real-world hindcasts when compared against field observations, demonstrating potential utility in applying such a
hierarchical analysis for model development applications.

Magnitudes of the northerly winds are enhanced by up to 0.5 m/s below the height of the jet and reduced by up to 0.6 m/s
above it in PM-O compared to ED-O, leading to a larger vertical wind shear (see Figure 1c). In PM-O, $\overline{v'w'}$ is also about half
as negative at altitudes between 300 m and 2 km, more in line with CM1. Differences in wind component structure between
ED-O and PM-O are related to differences in $\overline{u'_h w'}$ profile structure. Although the overall biases in $v$ are similar between ED-O
and PM-O, the differences in profile structure are very similar to those differences in the $v$ component profiles described by
Figure 8 in L19. L19 also found that their model predictions of both $v$ and $\overline{v'w'}$ profiles better matched LES when prognostic
$\overline{u'_h w'}$ were included in their model.

The PM-O simulation is also in better agreement, qualitatively, than ED-O, with the winds and $\overline{u'_h w'}$ profiles for BOMEX
found in Fig. 2 of D20. The smaller negative $\overline{u'w'}$ in the layer above the jet is closer to the nearly zero $\overline{u'w'}$ in this layer in
D20. Similarly, the less negative $\overline{v'w'}$ in the layer around the jet maximum in PM-O is more similar to the relatively weak $\overline{v'w'}$
in that layer in D20, although $v$ winds appear overall much weaker in BOMEX (around 1 m/s maximum) than in our study
(around 2 m/s maximum). These qualitative similarities to D20 in profiles predicted by PM-O make sense given that D20 also
noted the existence of countergradient fluxes in their simulations. These fluxes can be captured by the PM-O simulation but
not by ED-O because of the addition of prognostic $\overline{u'_h w'}$ calculations.

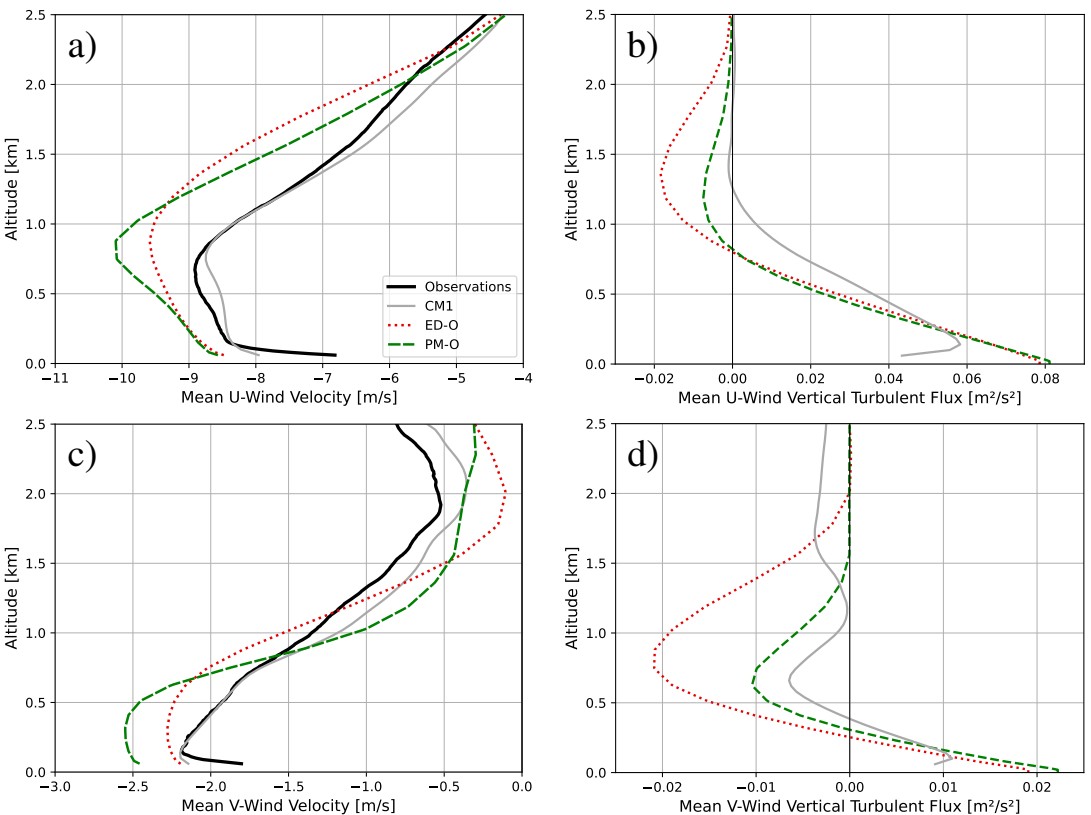

**Figure 1.** Domain mean vertical profiles from observations, CM1, ED-O, and PM-O for horizontal wind components ($u$ and $v$, panels a. and c.) and vertical turbulent fluxes of horizontal momentum ($\overline{v_h' w'}$, panels b. and d.).

Figure 2 displays mean profiles of both $u$ and $v$, and horizontal wind speed ($\overline{|U_h|}$) for both CAM configurations, along with corresponding vertical profiles of the biases and root mean squared errors associated with these variables. For both the bias (middle row) and RMSE (lower row), values closer to zero are desirable and reflect better agreement with the sounding data. As implied by Figs. 1a,b (reproduced as the top row of Fig. 2), it can be seen that although PM-O has a stronger jet maximum than ED-O, it has a reduced maximum easterly bias when compared to ED-O since its jet placement matches observations better (Fig. 2d). It can also be seen that the reduced easterly bias in PM-O corresponds with a reduced overall $\overline{|U_h|}$ bias (Fig. 2f). There is a noticeable decrease in the RMSEs of u and $\overline{|U_h|}$ of about 0.3 m/s moving from ED-O to PM-O in the region immediately above the modeled jet maximum (roughly 1 to 2 km altitude) (Figs. 2g,i). Both the RMSE profile for $v$ and the RMSE profile for $u$ far from the modeled/observed jet maxima are quite similar between ED-O and PM-O, which implies that other model biases are important drivers in solution error rather than $\overline{u_h' w'}$.

Upgradient fluxes are not apparent in any mean momentum profile in Fig. 1 as vertical wind shear ($\frac{\partial u_h}{\partial z}$) sign changes occur at nearly the same altitudes where $\overline{u_h' w'}$ sign changes occur in both ED-O and PM-O (although not exactly because of linear interpolations working on model levels of inconsistent heights). Upgradient fluxes are, however, present in individual

profiles. One way to describe where upgradient fluxes are occurring is by calculating an "effective eddy diffusivity" ($K_{eff}$) and finding where it is negative. This quantity backs out what the transfer coefficient $K_m$ described in Eqs. 1 and 2 would have to be in order to predict the given $\overline{u_h' w'}$ profile from the vertical wind shear. Eq. 8 describes this calculation essentially as a rearrangement of Eqs. 1 and 2. The coefficient $K_m$ is always positive in a model that diagnoses momentum flux (and thus $\overline{u_h' w'}$ always works downgradient). Here, a negative value of $K_{eff}$ indicates that upgradient fluxes are occurring.

$$K_{eff} = -\frac{\overline{u_h' w'}}{\frac{\partial \overline{u_h}}{\partial z}} \tag{8}$$

Figure 3 describes all model levels below 600 hPa on each of the 1,546 recreated soundings (before linear interpolation is applied) where negative values of $K_{eff}$ are found for $u$ for both ED-O and PM-O as black points. For ease of analysis, we are only concerned with the zonal components of wind shear and momentum flux here, although a cursory analysis of the meridional component showed similar results. Some points in ED-O are found to have negative $K_{eff}$, but these arise because CAM outputs $u$ and $\overline{u'w'}$ at different points within its timestep. This can lead to $u$ in low shear environments being updated by other subroutines such that small changes induce a sign flip in $\frac{\partial \overline{u}}{\partial z}$ which results in $K_{eff}$ being erroneously calculated as negative. In order to exclude such occurrences, points where $K_{eff}$ is found to be negative, but the absolute value of $\frac{\partial \overline{u}}{\partial z}$ is smaller than 0.15 m/s per km (i.e., essentially unsheared layers), are shown in orange. This threshold was chosen to be larger than the largest value of $\frac{\partial \overline{u}}{\partial z}$ found for any point with negative $K_{eff}$ in ED-O since this model configuration is incapable of generating true upgradient fluxes within the CLUBB subroutine. This removes between 0.1 and 0.2% of the points in either simulation. Most points with negative $K_{eff}$ in PM-O are above this threshold and remain black in the corresponding panel. It is evident that PM-O does indeed produce countergradient fluxes that are not apparent in the ED-O simulations.

Most upgradient $\overline{u'w'}$ predicted by PM-O fall in a layer between 950 hPa and 850 hPa, which roughly corresponds to 600 to 1400 m above the ocean surface. The CM1 EUREC[4]A/ATOMIC LES simulation performed here prognosed a layer of counter-gradient momentum fluxes between 925 hPa and 900 hPa (820 to 1060 m). These are similar ranges of altitudes as where L19 found upgradient fluxes when running the BOMEX test case with both LES and single-column models (their Fig. 1), which was between approximately 770 and 1070 m altitude. This layer is also approximately where Helfer et al. (2021) calculated nega-tive $K_{eff}$ between approximately 600 and 1700 m altitude for their large eddy model hindcast using data from the NARVAL campaign (their Fig. 10). This demonstrates that a high-order turbulence scheme can reproduce these countergradient fluxes in global ESM simulations and that they occur when the atmospheric state is initialized with real-world conditions. From these results and the $\overline{u_h' w'}$ profile structures of past LES, we speculate that the zonal jet is more physically realistically represented when prognostic $\overline{u_h' w'}$ is applied in lieu of traditional eddy diffusivity by comparing short-term initialized hindcasts using a cli-mate model compared to intensive field campaign data. Confidence is added to this hypothesis by qualitatively similar findings in recent work investigating LES simulations with atmospheric forcing consistent with EUREC[4]A/ATOMIC field campaign conditions. This underscores the utility of applying initialized hindcasts to help bridge the gap that has traditionally existed between process-oriented analyses (e.g., single-column models, LES, observations) and long-term (e.g., multi-decadal) climate simulations.

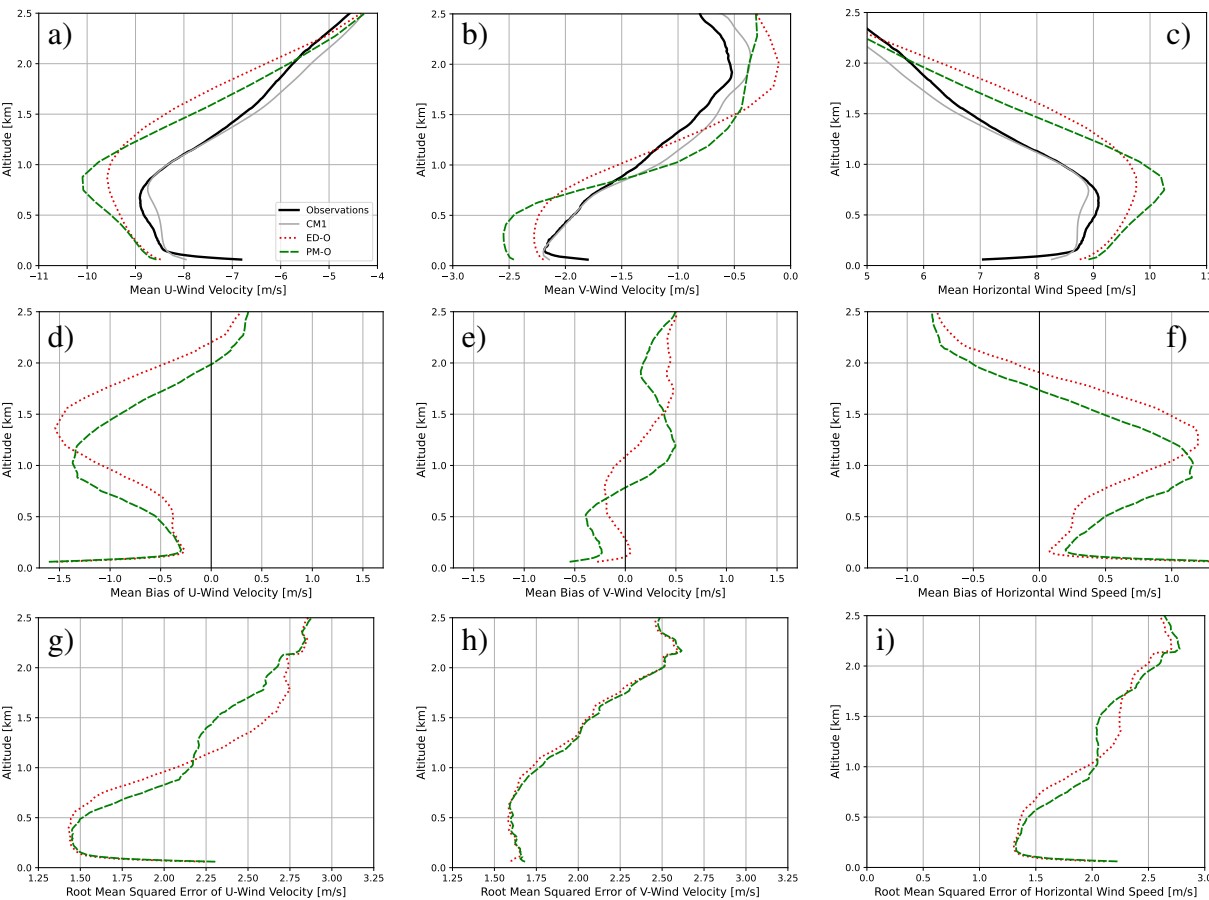

**Figure 2.** Vertical profiles of means (panels a.-c.), mean errors (biases) (panels d.-f.), and root mean squared errors (panels g.-i.) of various CAM configurations for horizontal wind components $u$ (left column) and $v$ (center column) and overall horizontal wind magnitude $|U_h|$ (right column). Observations are also included in the mean profiles. Note that panels a. and b. are reproduced from Fig. 1a,c.

### 3.2 Thermodynamic Profiles

While we only change equations related to $\overline{u'_h w'}$ in PM-O, it is worth considering how these changes may feed back onto the atmospheric state and therefore modulate thermodynamic profiles ($T$ and $Q$) and their fluxes. It is revealed in Fig. 4 how the predictions of thermodynamic quantities also change when the prognostic $\overline{u'_h w'}$ formulation is introduced. Figure 4a displays profiles of $\theta$ rather than $T$ itself to highlight the stability of layers. The default run features a sizeable cold bias for all altitudes below 2.5 km, a cold bias that is only slightly changed (on the order of a tenth of a Kelvin) in PM-O. In observations, the domain mean $Q$ profile features a "dry nose" around 1 km and a "moist nose" around 1.7 km while both model configurations predict a smoother decrease in moisture with height, meaning they both have moist biases around 1 km and dry biases around 1.7 km altitude. Both configurations also have a dry bias in the lowest 500 meters. Although the directions of these biases are consistent between model configurations, their magnitudes do change on the order of a few tenths of g/kg. The dry bias below

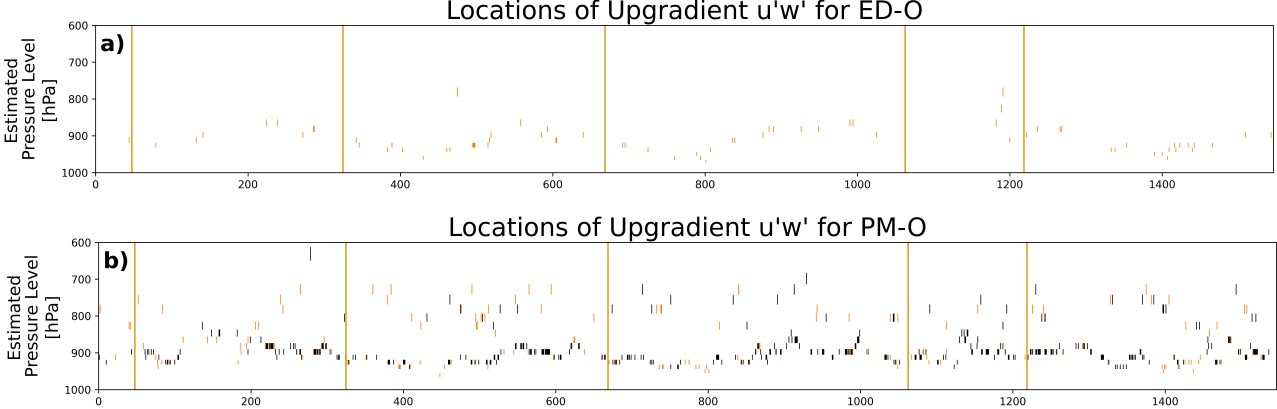

**Figure 3.** Diagrams displaying where effective eddy diffusivity ($K_{eff}$) is negative (and thus where upgradient fluxes are occurring) in ED-O and PM-O output for the vertical flux of zonal momentum ($\overline{u'w'}$). Black and light red dots indicate where upgradient fluxes are calculated to be occurring, but light red dots indicate where negative $K_{eff}$ was also calculated with a very small value for the vertical gradient of zonal momentum ($\frac{\partial u}{\partial z} < 0.15 \frac{m}{s}$ per km) (and thus where the upgradient flux calculation is likely spurious). The vertical axis is a rough estimate of the pressure level of the model output and the horizontal axis is the index of each re-created sounding in the original data. Pressure levels here are taken from a column at a single time, making the pressure levels estimates, since the hybrid pressure coordinates change depending on elevation and surface pressure. In this situation, this is a reasonable estimate since all balloons were launched from near sea level and almost all drifted over the open ocean in fair weather conditions. The vertical yellow lines separate the soundings based on which observatory or "mission" they are from. Within each mission, the soundings are in chronological order. From left to right, the six "missions" are those balloons launched from L'Atalante with Meteomodem radiosondes, L'Atalante with Vaisala radiosondes, the Barbados Cloud Observatory, Meteor, Maria S. Merian, and the Ronald H. Brown.

500 m is roughly cut in half from about 0.4 g/kg in ED-O to 0.2 k/kg in PM-O while the dry bias centered around 1.7 km is degraded in PM-O by around 0.1 g/kg.

These differences in thermodynamic profiles are not as large as the differences in the momentum profiles but do exist. In fact, these differences are still significant at most altitudes when performing a paired Student's $t$-test across the model profiles included in Fig. 4 (92% (72%) of altitude bins in the $\theta$ ($Q$) profiles significantly differ between ED-O and PM-O at the $\alpha = 0.05$ level). This would seem to contradict the findings in L19 where there was no noticeable difference found in the thermodynamic profiles predicted by the prognostic versus the diagnostic $\overline{u'_h w'}$ configurations of the single-column model. The structures of the thermodynamic profiles from the LES in L19 are very similar to those from observations in this study, and those profiles from the single-column model in L19 have similar shapes to the CAM output in this study. We hypothesize that the differences in thermodynamic profiles between ED-O and PM-O indicate there is additional two-way feedback between $\overline{u'_h w'}$ and scalar fluxes in CAM due to the hindcast framework (i.e., $\overline{u'_h w'}$ changes the atmospheric state, which is further modified and advected by the dynamical core, which then is passed back to the physical parameterizations, including CLUBB, etc.). This feedback

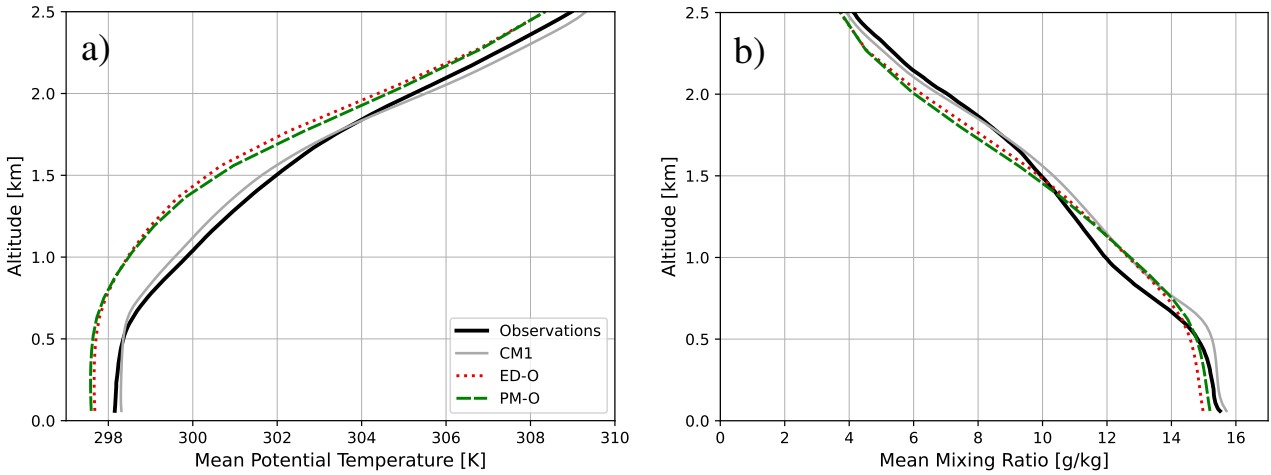

**Figure 4.** Domain mean vertical profiles from observations, CM1, ED-O, and PM-O for potential temperature ($\theta$) and water vapor mixing ratio ($Q$).

would not occur in the single-column model in L19 (which applies a large-scale nudging to specify the mean state fields that are used by the subgrid turbulence scheme).

## 4 Results of the Experimental Vertical Turbulent Length Scale Formulation

### 4.1 Dynamic and Thermodynamic Profiles

The impact of applying the experimental estimate of $L$ in simulations can be assessed using the ED-X and PM-X results. Recall that these runs either diagnose momentum fluxes via eddy diffusivity or prognose them directly as above but add an experimental modification to how $L$ is calculated. Results for these simulations are shown in Figs. 5 and 6, which are similar to Figs. 1 and 4 except they include the additional CAM configurations with the experimental formulation for $L$ using the coefficient values described in Section 2.3.

Like PM-O, both experimental length scale runs ED-X and PM-X have an easterly jet that is more narrow. Unlike PM-O, however, (which has an enhanced easterly wind bias at the jet maximum), PM-X features a reduced easterly bias relative to ED-O in this layer (Fig. 5a). Profiles of $v$ in PM-X also tend to qualitatively match observations better than PM-O (Fig. 5c). Both ED-X and PM-X produce $\theta$ profiles with cold biases a few more tenths of Kelvins smaller than both PM-O and ED-O (particularly near and just above the jet) and $Q$ profiles that match observations more closely than both PM-O and ED-O at most altitudes (Figs. 6a,b). The dry bias in the lowest 500 meters is nearly eliminated in PM-X (Figs. 6d).

How simulated wind biases depend on the time of day is described for all model configurations in Fig. 7 (based on sounding launch local time). On plots corresponding to $u$ and $v$ components, red colors indicate where CAM tends to predict values that are too negative (more easterly or more northerly) than in reality, while blue colors indicate where wind components are too

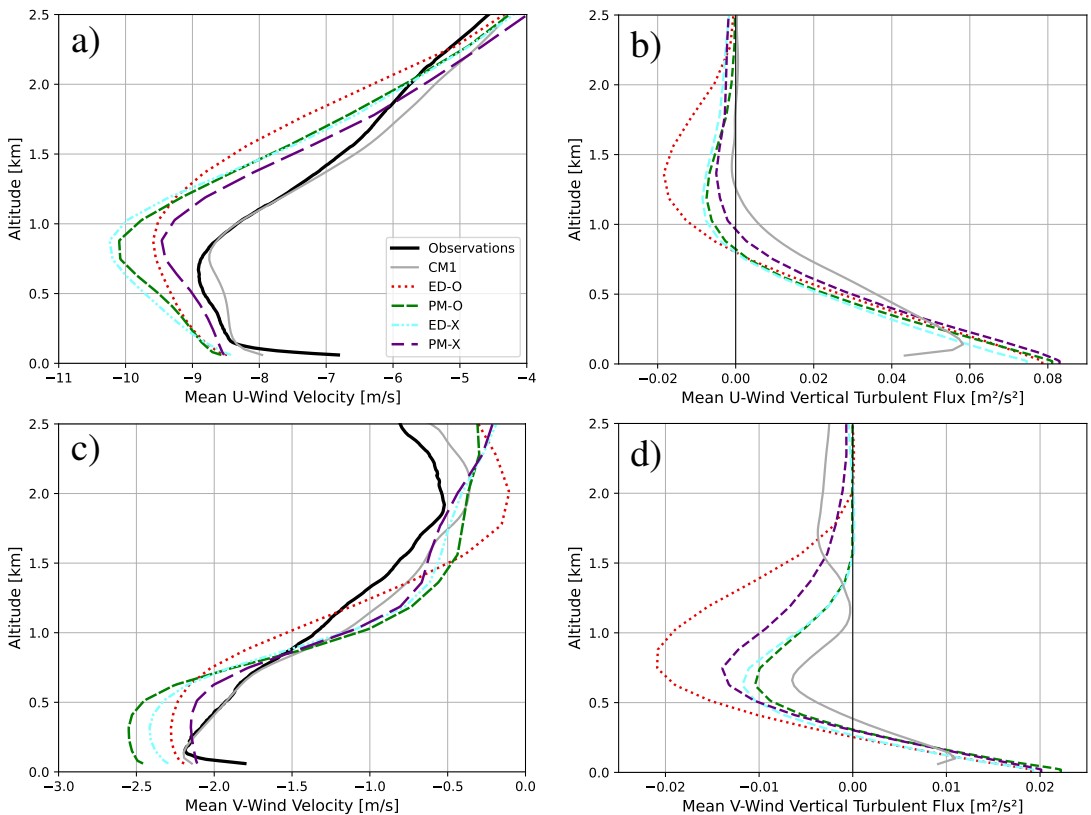

**Figure 5.** As in Fig. 1 but including all CAM configurations

positive. On the plots of $\overline{|\boldsymbol{U_h}|}$, the violet colors indicate where CAM tends to overpredict the magnitude of the wind, while green is where it tends to underpredict. The jet layer easterly (negative) bias in the default run is present at all times of day, but strongest in the daytime. A smaller magnitude westerly (positive) bias seems to exist between 2 and 5 km in ED-O: present at most times of day, except the afternoon when it is small or slightly reversed. Much like the wind magnitudes themselves, biases in $v$ are generally smaller than those of $u$, but generally, ED-O features a background southerly (positive) bias that is

largest at night and away from the surface. Bias in $\overline{|\boldsymbol{U_h}|}$ appears dominated by biases in $u$, with winds being too strong in the jet layer, especially in the daytime, and too weak above this, especially at night.

Bias reduction in $u$ when adding the prognostic $\overline{u_h'w'}$ equation can be seen here at almost all times of day when moving from ED-O to PM-O (Figs. 7a,d), particularly between about 1 and 2 km altitude where the maximum magnitude bias changes from around -1.5 m/s to around -1.2 m/s. Moving to the experimental length scale runs, the ED-X $u$ bias (Figs. 7g) is generally larger

than PM-O. However, the bias is minimized in these runs when combining both prognostic momentum and the experimental length scale (PM-X), especially in the lowest 2 km where the maximum magnitude bias becomes around only -1.0 m/s (Fig. 7j). For $v$ wind, biases appear mostly the same in all configurations (Figs. 7b,e,h,k) with perhaps the background nocturnal southerly bias being made a few tenths of a m/s worse in the experimental length scale runs. Biases in $\overline{|\boldsymbol{U_h}|}$ are similarly

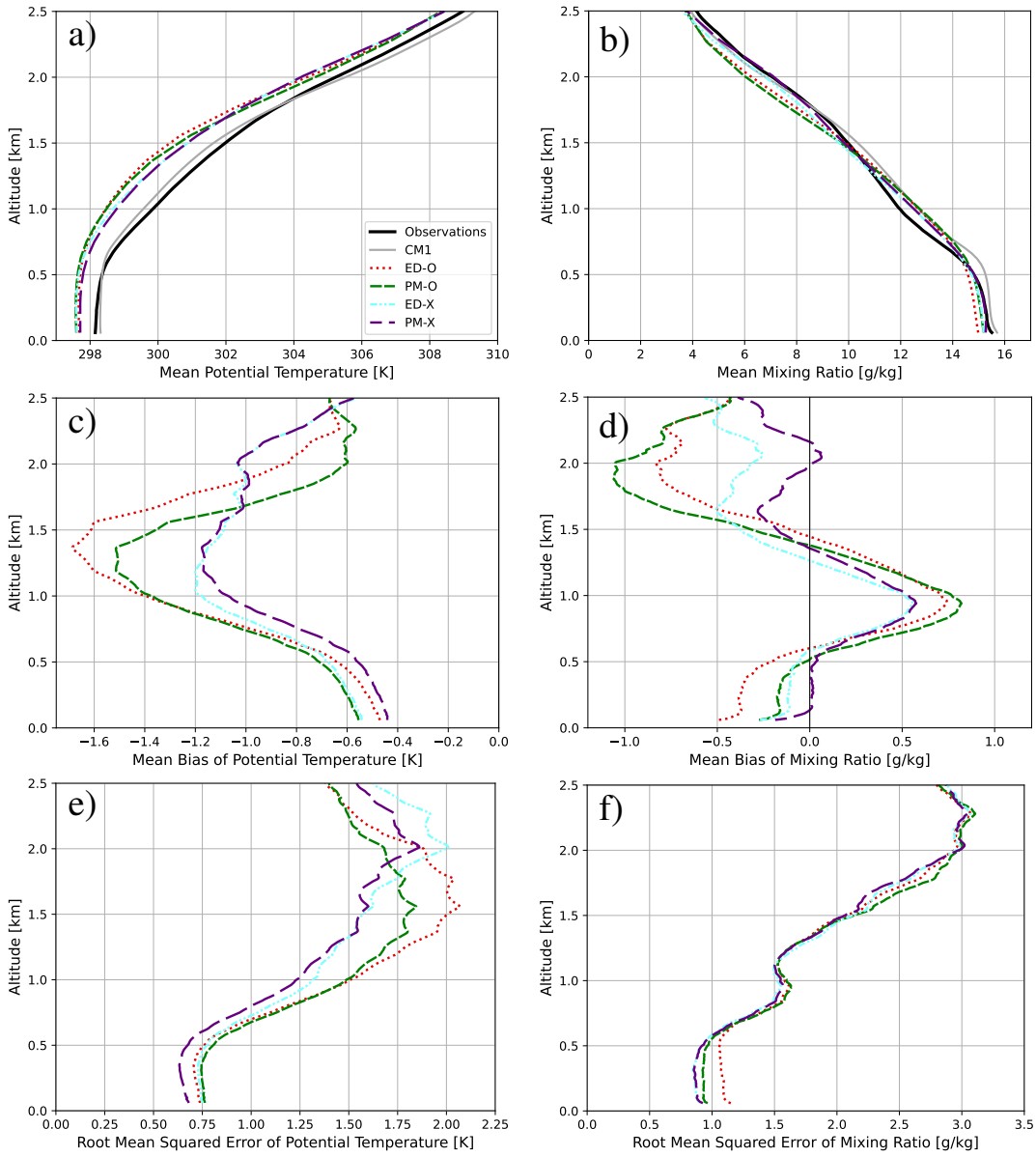

**Figure 6.** Vertical Profiles of domain mean (top row), mean bias (middle row), and root mean squared error (bottom row) of potential temperature ($\theta$) (left column) and water vapor mixing ratio ($Q$) (right column) for all CAM configurations.

reduced moving from ED-O (Fig. 7c) to PM-O (Fig. 7f) and further reduced moving from PM-O to PM-X (Fig. 7l), likely

owing to the dominance of $u$ biases.

Errors in state fields throughout the rest of the troposphere (above 2.5km) are largely unaffected by the differences between CAM configurations (not shown). Consistent biases in the background tropospheric likely arise from errors in model initialization and from other effects such as discretization in the dynamical core and the subgrid parameterization of other processes. Such errors will propagate into boundary layer prediction no matter the skill of the turbulence parameterization, particularly

when one considers the free atmosphere as an upper boundary condition to the system. Along with errors arising from the surface layer formulation, these are likely why the general pattern of bias sign with regards to altitude and time of day remains quite similar for all configurations despite improvements seen in bias magnitude for boundary layer winds.

Diurnal cycles of mean biases for these three momentum variables between 200 m and 2 km are described in Fig. 8. This range of altitudes is chosen to focus on errors in the boundary layer and to exclude errors in the surface layer and the free

troposphere. Errors tend to saturate around 2 km in all model configurations, becoming constant with height (e.g., Figs. 2g-i). There is a clear pattern in observations where the winds tend to be weakest in the early afternoon and strongest in the early morning hours. This mean diurnal cycle is captured in each model configuration, but the magnitude of the easterly wind component is always overpredicted. All 3 panels have a range of 3 m/s on their vertical axes. A minor mean reduction in the strong easterly jet bias of around 0.1 m/s can be seen moving from ED-O to PM-O in Fig. 8a. The addition of the experimental

length scale with the eddy diffusivity code (ED-X) either slightly increases or slightly decreases error (relative to ED-O) depending on time of day. However, much greater mean bias reductions in the range of 0.2 to 0.4 m/s can then be seen by combining both updates in PM-X. By comparison, biases in $v$ are all very small, making the mean bias patterns for $\overline{|U_h|}$ essentially the same as those in $u$ (except a more negative $u$ is a larger $\overline{|U_h|}$ here).

Figure 9 describes where negative values of $K_{eff}$ are found for $u$ for the experimental length scale runs alongside PM-O

in the same manner as Fig. 3. Like ED-O, no true countergradient fluxes are observed in ED-X, an expected result given the assumption of downgradient diffusion. Upgradient fluxes are more common and tend to occur in deeper layers in PM-X compared to PM-O, although they are still most common in the layer from 950 to 850 hPa, they now often extend higher to near 750 hPa (or roughly 2500 m). We emphasize that these more frequent predictions of upgradient fluxes are not necessarily more accurate, however, they do demonstrate a likely connection between the prediction of countergradient fluxes and modifications

to the turbulent dissipation in CLUBB. That is, in the 'PM' simulations, changes to the turbulent length scale aimed at improving the shape of the near-surface $u$ and $v$ profiles can further enhance the generation of upgradient momentum fluxes. Figure 10 shows a frequency distribution of the actual $K_{eff}$ values from Fig. 9 with the values under extremely low wind shear masked to remove interpolation artifacts as discussed earlier (between 0.1 and 0.2% of the values). The numeric value in the legend indicates the number of $K_{eff}$ estimates less than 0, indicating countergradient fluxes (i.e., the fractional occurrence of black

points in Figs. 3 and 9). No countergradient fluxes are indicated for the eddy diffusivity (ED) runs, although 1.2% and 5.9% of zonal momentum fluxes are countergradient in the PM-O and PM-X simulations, respectively.

While the specific focus of this work is on the transport of momentum, we show vertical profiles of cloud liquid and cloud fraction in Fig. 11 since a key motivation for understanding boundary layer processes in this region is to improve the

representation of low clouds in Earth system models (and their associated forcing on the climate system). When prognostic momentum is turned on (ED-O to PM-O) both cloud liquid and cloud fraction decrease. A decrease in the height of peak cloudiness also occurs. Both of these changes tend to represent a better agreement with the CM1 LES results, although we stress that we have not undertaken a rigorous comparison with observations from a cloud perspective. Nonetheless, we do note these results are qualitatively similar to those published in Narenpitak et al. (2021) and Schulz and Stevens (2023). Turning on the experimental length scale formulation (ED-X and PM-X) results in an increase in cloud liquid and a further reduction in the height of the peak cloudy layer. Both of these further improve the correspondence of the profile shape to the CM1 results, although both liquid and fraction are overestimated in magnitude relative to the LES run. Somewhat interestingly, going from ED-X to PM-X increases cloud liquid, which is counter to the same change using the original length scale formulation (ED-O to PM-O). While this is just a cursory look at cloud fields, it would imply that changes in the treatment of momentum fluxes also feed back into cloud fields, but that the updated treatment of $\tau$ may play an equally or larger role. This is unsurprising given that $\tau$ appears in many equations throughout CLUBB, not just those associated with momentum (Golaz et al., 2002). These cloud responses to both momentum treatment and length scale formulation are complex and merit additional evaluation and calibration.

We conclude this section by pointing out that these experimental length scale runs should be treated more akin to a sensitivity analyses. In other words, we explore how more generalized treatments of eddy turnover timescales could impact simulated state profiles when coupled to two different momentum flux treatments in the study region. Given how PM-X appears better at reducing biases in thermodynamic fields than PM-O, it may be useful to pursue more formalized tuning processes (i.e., beyond the Nelder-Mead method applied here) in future work.

### 4.2 Mean Biases and Root Mean Squared Errors

To quantify the performance of these configurations in simulating EUREC$^4$A/ATOMIC observations, Figure 12 displays the mean biases between altitudes of 200 m and 2 km for each CAM configuration in several state variables. Biases are first calculated for each sounding profile and then the mean is taken over all soundings at each specific altitude (every 10 meters). The blue and red shadings indicate how these biases have changed from the default run (ED-O). Red colors indicate that the absolute magnitude of the mean bias has increased and blue colors indicate that this magnitude has decreased. The color scale here runs from a 100% decrease in bias magnitude in the darkest blue (complete bias elimination) to a 100% increase in the darkest red (doubling of the bias).

Starting on the left, the column for ED-O is completely white because each value serves as the reference bias for the corresponding variable. When the prognostic $\overline{u_h'w'}$ is added in PM-O, mean biases are reduced on the order of 5 to 10% for most variables. The exceptions to this are $v$ and $Q$, which see very slight increases in mean bias. The coloring here is not particularly meaningful for these two variables, however, given how small the corresponding mean biases are in ED-O to begin with (a minuscule absolute change in these biases appears as a significant relative change). The fact that the $\overline{|U_h|}$ bias is reduced also implies that the $u$ bias reduction is a more important contributor than the $v$ bias degradation. Moving now to the third column with the experimental length scale with eddy diffusivity (ED-X), the picture is similar. There is less

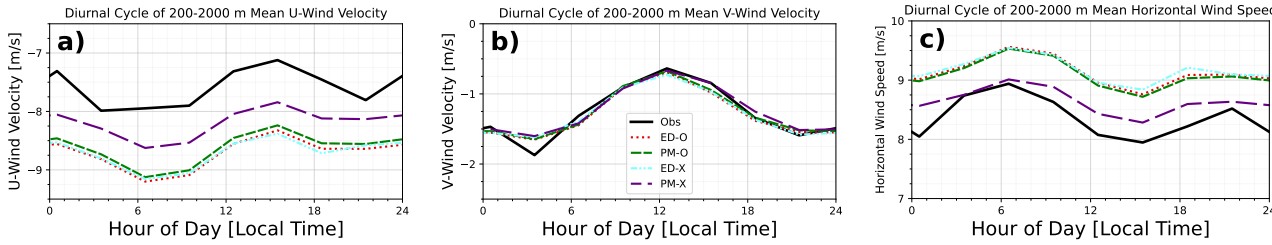

**Figure 7.** Plots of biases in mean zonal wind speeds ($u$) (left column), meridional wind speeds ($v$) (middle column), and horizontal wind magnitudes ($\overline{|U_h|}$) (right column) as a function of time of day and altitude predicted by runs ED-O (a,b,c), PM-O (d,e,f), ED-X (g,h,i), and PM-X (j,k,l). Biases are averaged every 10 meters of altitude and in eight 3-hour blocks based off of sounding launch times.

**Figure 8.** Mean biases in mean zonal wind speeds ($u$) (a), meridional wind speeds ($v$) (b), and horizontal wind magnitudes ($\overline{|U_h|}$) (c) predicted by each CAM configuration averaged between 150 m and 750 m altitude. Biases are averaged in eight 3-hour blocks based off of sounding launch times.

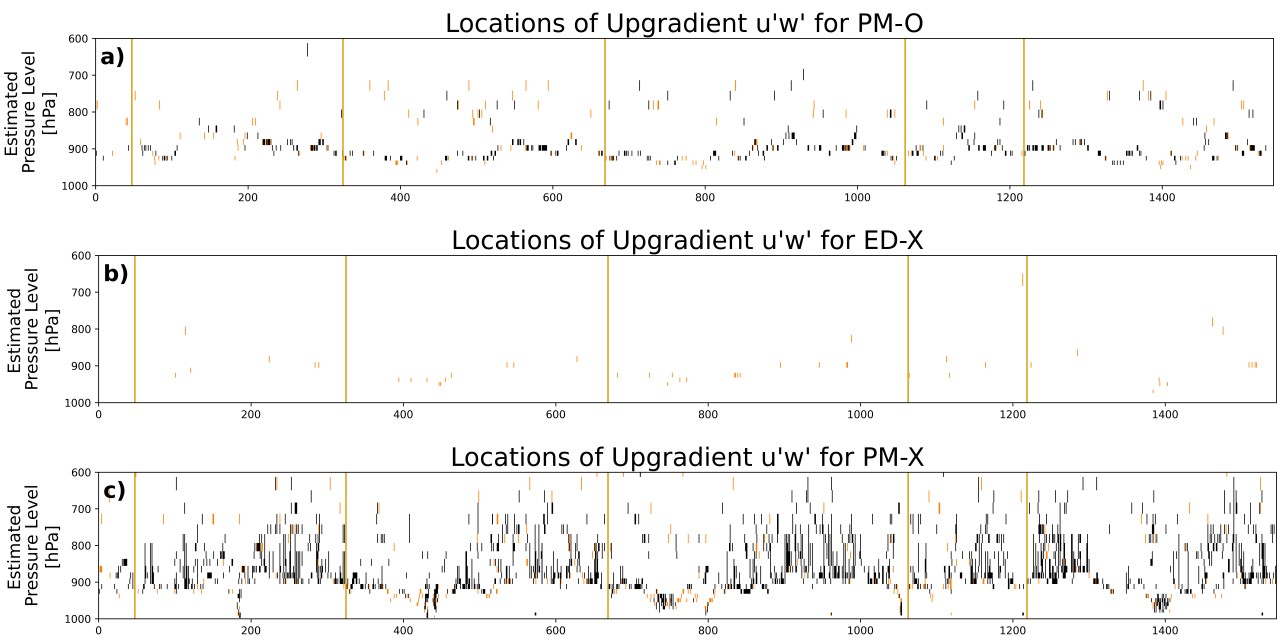

**Figure 9.** As in Fig. 3, except for CAM output from PM-O, ED-X, and PM-X. Note that PM-O (panel a) in this figure is identical to panel b) in Fig. 3).

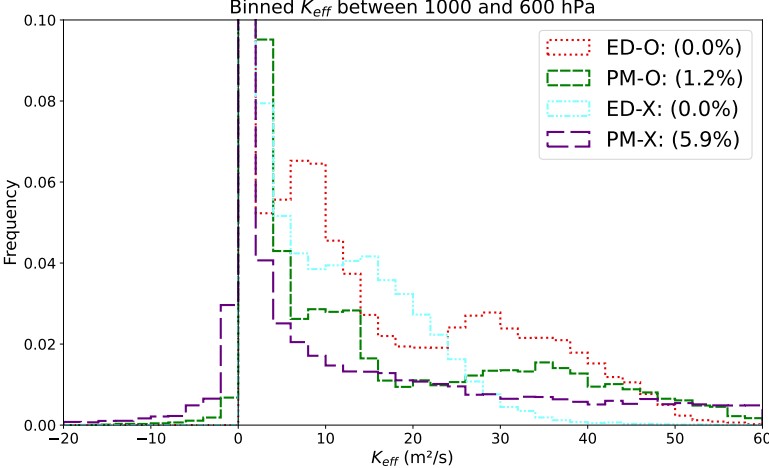

**Figure 10.** Histogram of $K_{eff}$ values for each of the profiles in Figs. 3 and 9. Bin widths are 2 m$^2$ s$^{-1}$.

(more) improvement from a bias perspective in the momentum (thermodynamic) quantities, although these differences aren't overly large. The final column includes both changes to the code (PM-X) and represents some combination of the second and third columns. In this column, the blue shading becomes darker, indicating a further reduction of mean bias in most variables.


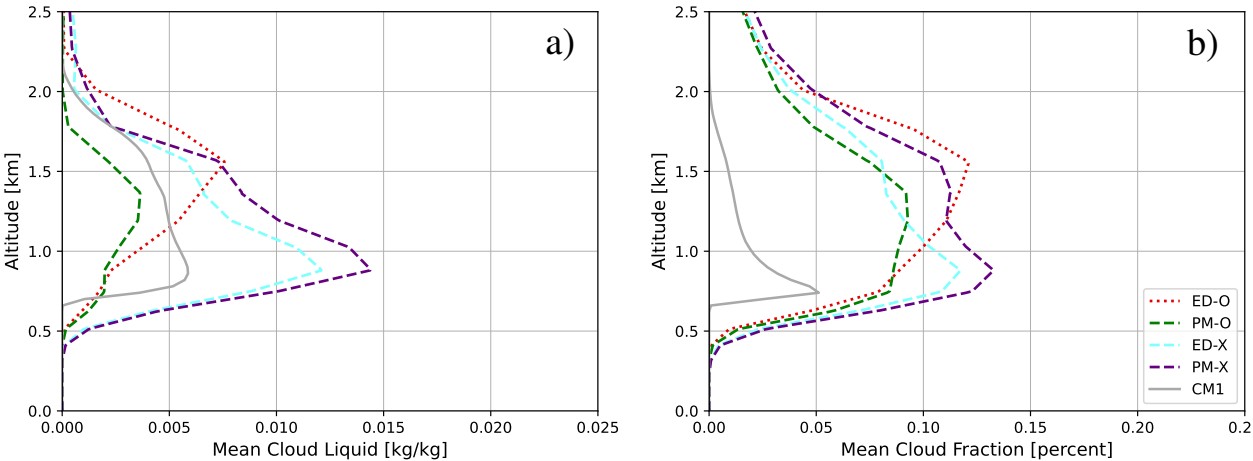

**Figure 11.** Domain mean vertical profiles from observations, CM1, and CAM simulations for mean cloud liquid and mean cloud fraction.

The greatest improvements are seen in $u$ and $\overline{|U_h|}$, as was seen in the profiles with the better depiction of the jet. Some bias degradation is seen in these means for $v$ and $q$. However, we also emphasize that these results are not overly meaningful since the mean biases for both these variables are small to begin with and therefore the absolute changes in biases between model configurations are small as well (even if the ratio that governs the shaded underlay is large).

Biases cannot paint a full picture, since they do not account for errors that have no mean tendency. Figure 13, is identical to Fig. 12 except it describes root mean squared errors (RMSEs) rather than biases (and has a much more sensitive color scale that runs from a 15% decrease to a 15% increase). Predictions of $u$ are indeed improved when measured by aggregate RMSE reduction (albeit by a few percent) in PM-O. Although mean $u$ bias between 200 m and 2 km is reduced in PM-O relative to ED-O, recall that the improvement in the structure of the wind profile seen when moving from ED-O to PM-O is accompanied

by an increase in the strength of the easterly jet, which itself has an easterly bias in ED-O (see Fig. 2). The worsened $u$ biases at certain altitudes in PM-O likely counteract any improvements in layer-mean RMSEs that may come from a more accurate wind profile structure. Improvements in thermodynamic fields are also visible as reductions in RMSEs. This is particularly interesting for PM-O relative to ED-O since the code used to calculate the turbulent fluxes of scalars (i.e., $T$ and $Q$) was the same in these runs. Such improvements again suggest downstream effects of better resolving momentum profile structure via

feedback with mean state fields: a phenomenon not seen in single-column models.

     The ED-X simulations include larger reductions in RMSE for $T$ and the closely related $\theta$ – ranging between 10 and 20 percent – although larger degradations in the wind profiles when compared to PM-O. These apparent temperature improvements are likely dominated by the reduction of the cold bias seen at almost all altitudes when moving from ED-O to ED-X. A correspondence of those altitudes with the greatest cold bias reduction to those altitudes with the greatest RMSE reduction can

be seen in Fig. 6. Combining the two updates (PM-X) results in RMSE improvements for each variable when compared to ED-O, implying that a combination of the prognostic momentum and the experimental length scale improves the simulation fidelity. This provides further evidence for both of these modifications to jointly improve boundary layer structure and for

the significance of a two-way dynamic-thermodynamic feedback. The results of the PM-O and ED-X runs imply that the prognostic momentum is a larger player in reducing errors associated with winds over the EUREC[4]A/ATOMIC region, with the experimental length scale and associated parameter settings further reducing the RMSE improvements seen with just the prognostic momentum alone.

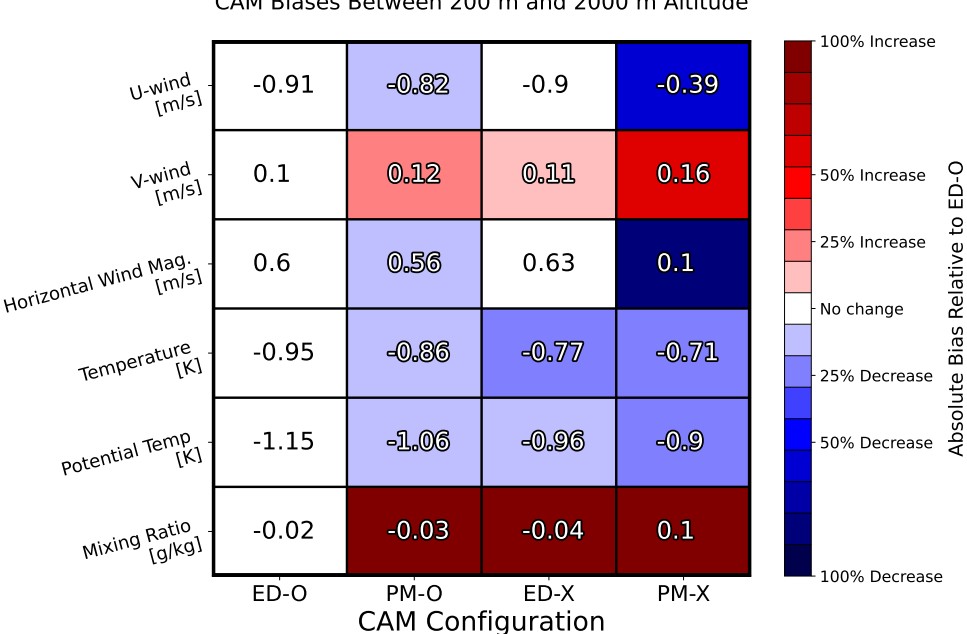

**Figure 12.** Chart describing absolute errors (biases) of CAM predictions between 200 m and 2 km altitude relative to sounding data for all model configurations and state variables. All levels are equally weighted. Numbers in each cell describe the actual bias for the corresponding variable and configuration. Colors describe how these errors compare to that of the same variable in the default configuration (ED-O). Blue colors indicate that the error has a smaller absolute value and red colors indicate that the error has a larger absolute value compared to ED-O. The colors are scaled such that the darkest blue would be a complete bias eradication and the darkest red would be a doubling of the reference bias in ED-O. All levels are equally weighted.

### 4.3 Horizontal Momentum Budgets

Given the improvement in wind profile predictions relative to observations moving from PM-O to PM-X, it is worth comparing how the individual terms that contribute to the time tendency of $\overline{u'_h w'}$ in Eq. 3 differ between them. If the state variable predictions of a given configuration better match observations, it is conceivable that the corresponding modeled momentum budget profiles that helped make these predictions are themselves better descriptions of physical reality. In other words, studying these budget terms may provide physical insight into why one configuration's predictions may be better than those of another. Note that only the simulations with prognostic momentum produce a budget to analyze, hence there are no budgets for ED-O and ED-X. Figure 14 describes vertical profiles of the $\overline{u'_h w'}$ budget terms described in Eq. 3 for both PM-O and PM-X.

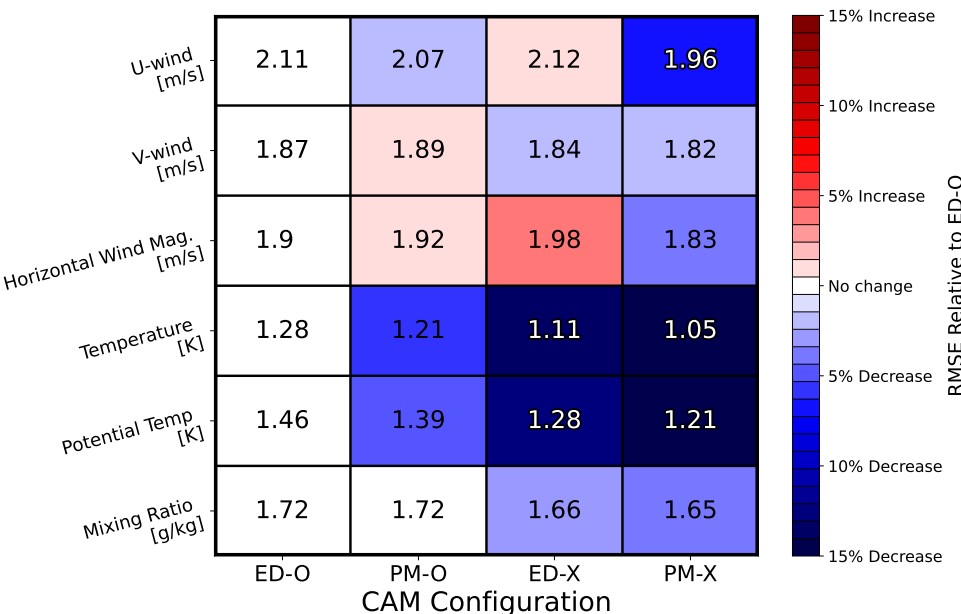

**Figure 13.** As in Figure 12 but for root mean squared errors. Notice that the color bar has changed to have extrema of +/- 15%.

The "mean advection" term (1) corresponds to the advection of existing $\overline{u'_h w'}$ by the mean vertical wind, while the "turbulent advection" term (2) represents that advection by turbulent perturbations in $w$. Term 3 is the turbulent production of $\overline{u'_h w'}$ by variance in $w$ acting in a vertical $u$ or $v$ gradient, while term 4 is that turbulent production by pre-existing $\overline{u'_h w'}$ acting in a vertical $w$ gradient.

The "buoyant production" term (5) describes the net change in $\overline{u'_h w'}$ from covariance between parcels of particular values of buoyancy and with horizontal momentum. "Return to isotropy" (term 6) refers to the effective dissipation of $\overline{u'_h w'}$ determined by $\tau$, and term 7 is the "residual" dissipation term. The "time tendency" is the sum of all other terms (the left-hand side of the equation), but here, this term is typically orders of magnitude smaller than any of the individual budget terms because of how many terms nearly balance each other. In order to make the overall time tendency apparent on the same x-axis scale, it is multiplied by 10 in Fig. 14.

One of the most notable differences in these plots is the strong reduction of turbulent production (by $\overline{w'^2}$) in the lowest 1 km in PM-X compared to PM-O for both the zonal and meridional components (solid brown lines in Fig. 14). This is accompanied by a similar reduction in the compensating "return-to-isotropy" term (dotted purple lines), whose magnitude is related to the magnitude of the net $\overline{u'_h w'}$ produced. Another notable difference is the changing of the sign of the buoyancy production term (solid blue lines) from weakly negative in PM-O to notable positive below 700 m, and negative above in PM-X, particularly in the zonal momentum budget. This is also qualitatively consistent with the BOMEX LES budgets in L19 (their Fig. 7) which lend credence to process-level improvement in the PM-X runs. We hypothesize that this may be related to increased

stratification in the $\theta$ profile in PM-X making vertical transport or air parcels due to buoyancy more difficult in the lowest 700 m. In that case, improvement in the thermodynamic profile in PM-X could be leading to changes in atmospheric stability (e.g., note the differences in the change in $\theta$ with height in Fig. 6a), which in turn lead to changes in buoyant production of $\overline{u_h'w'}$ which then feeds back to changes in the dynamic profiles. Since downgradient diffusion corresponds to a simple balance between turbulent production and return-to-isotropy, the fact that the buoyancy term is so large in PM-X could explain the enhanced upgradient fluxes in Fig. 9. We admit this is speculative, however, and experiments with more constrained model configurations (e.g., single column, nudged runs) and voluminous diagnostics would be helpful in providing deeper insight, including more detailed consideration of other turbulent quantities such as the vertical fluxes of temperature and moisture as well as variances (e.g., $\overline{u'^2}$ and $\overline{v'^2}$ would be directly affected by the additional of prognostic momentum to CLUBB).

While relatively qualitative in nature, the evaluation of initialized model simulations against observed state profiles, with subsequent analysis of turbulence budget terms that either improve or degrade said profiles could provide useful pathways for parameterization tuning and physical interpretation in future work. The lack of direct observations of turbulent quantities in this study limits the depth of analysis that can be done here. Estimating similar budgets from LES could prove useful in understanding whether these changes within the $\overline{u_h'w'}$ budget that lead to more skillful vertical provides are truly due to improvements in physical processes. This is a target for future work.

## 5    Discussion

We use 1-day-lead hindcasts produced by a general circulation model (CAM6) to evaluate its prediction of planetary boundary layer structure in a tropical maritime trade-wind regime. CAM is run in various configurations which vary in how turbulent momentum fluxes are calculated. A pair of configurations (ED) diagnoses these $\overline{u_h'w'}$ by implementing traditional downgradient diffusion while another pair of configurations prognose $\overline{u_h'w'}$ (PM) using the unified turbulence scheme, CLUBB. One of each momentum treatment uses the default calculation for a vertical turbulent length scale estimate included in CLUBB, while the other two use a more generalized equation to derive $L$ from the eddy diffusivity timescale, each with a different set of empirically determined coefficients. Predictions from each configuration are evaluated through comparisons to high-quality, high-resolution, real-world data from 1,546 weather balloons launched during the EUREC[4]A/ATOMIC field campaign.

Default CAM6 with standard eddy diffusivity (ED-O) is found to be too diffusive over the EUREC[4]A/ATOMIC domain. That is, when compared to observations, it predicts a jet that is too broad in terms of altitude and vertical gradients of $u$ and $v$ that are too weak. The introduction of prognostic $\overline{u_h'w'}$ reduces these biases by predicting a narrower jet, albeit one that is still too strong in terms of maximum velocity. This is a qualitative improvement in terms of how well the structure of this jet matches both observations from EUREC[4]A/ATOMIC and results from LES in both L19 and D20. This suggests higher-order momentum flux formulations, particularly those that permit countergradient fluxes, may be able to improve the representation of lower troposphere structure in trade-wind regimes, perhaps in conjunction with improvements to the surface layer formulation.

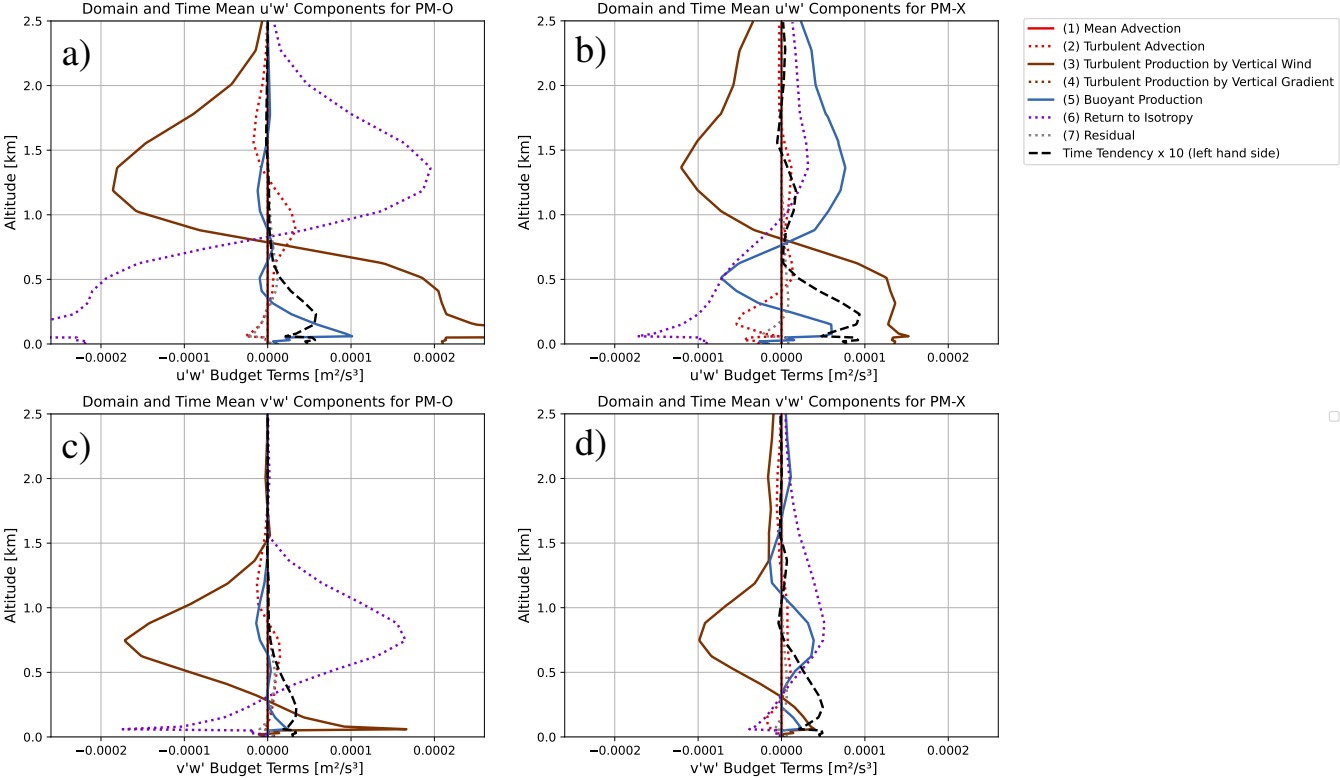

**Figure 14.** Domain-mean, time-mean, vertical profiles for the components affecting the time tendency of $\overline{u'w'}$ (top row) and $\overline{v'w'}$ (bottom row) for PM-O (left column) and PM-X (right column) as described in Eq. 3.

Further improvements in the prediction of boundary layer wind profiles are observed (as measured by root mean squared error in the relevant layer) when the experimental formulation of the turbulent length scale $L$ as first described in Guo et al. (2021) is included, and the relevant parameters are quasi-optimized. This suggests a more flexible, regime-specific strategy for estimating $L$ in GCMs can provide further improvement in the vertical structure of $\overline{u'_h w'}$ and subsequent wind profiles in these regions. These results do not point to any particular set of parameters leading to the best predictions but rather demonstrate that model predictions of boundary layer structure are sensitive to and can be improved via the tuning of these coefficients. While we only evaluated a subset of targeted dissipation permitted by this experimental length scale treatment, other possibilities, such as the additional damping of the third-order moment of vertical velocity in stable layers described in Guo et al. (2021), merit further study.

This study is a targeted regional investigation and as such, the improvements seen here cannot necessarily be generalized to the global climate system without further exploration. Reductions of errors in any particular run here do not necessarily imply that that run would generate better predictions globally. A parameterization that improves the structure of the boundary layer in a steady-state shallow cumulus regime over a relatively homogeneous calm ocean might also make predictions worse in regions with more orography and heterogeneous dynamical forcing. Model grid spacing is still on the order of $1°$ in mountainous

regions where topography can vary vertically by kilometers and thus these regions have the same requirement for subgrid parameterizations. How $\overline{u_h' w'}$ in the boundary layer responds to this roughness in boundary layer structure still must be captured by the same parameterization used by the model over the flat ocean surface. Therefore, one suggestion arising from this work is to more closely tie model development experiments to a variety of field campaign datasets and regions.

Although forecasts may improve when $\overline{u_h' w'}$ is prognosed rather than diagnosed, potential trade-offs exist in terms of computational cost and complexity. In the case of the CLUBB code specifically, the total computational cost of CLUBB increases by a few percent when adding prognostic $\overline{u_h' w'}$ if scalar fluxes have already been prognosed. This is only because many of the computations used to calculate scalar fluxes can be reused (Larson et al., 2019). Nonetheless, this is an increase in computational cost and one that would be larger in a model where a high-order closure is not already being implemented. Even besides the issue of computational cost, the inclusion of equations with more terms used to prognose $\overline{u_h' w'}$ increases the complexity of the model, thereby increasing the risk of introducing artifacts and increasing the difficulty of understanding model behavior (Mihailović et al., 2014).

The use of short-term initialized hindcasts here can serve to bridge a hierarchy gap between using long-term climate integrations and using single-column models or LES as tools for improving GCMs. This can be done since the large-scale environment is realistic in these hindcasts while significant model biases still appear within 1-2 days after initialization as can be seen here with CAM's prediction of wind speeds and jet height. Unlike in single-column models, here the simulated atmosphere can vary spatially and the subgrid parameterizations in question are allowed to impact the large-scale flow. This is unlike the 'one-way' transfer of information generally conferred by nudged configurations. Since the model is initialized with an observed state, observational profiles can be directly compared to the model simulation in a deterministic sense, rather than being averaged and compared to climatology in a more traditional assessment. Initialized simulations are also cheap compared to traditional climatology comparisons, with the four different sets of experiments here costing less than a single multi-year tuning run typically used by climate modeling centers.

Improvements in boundary layer structure are likely limited by the propagation of errors from the near-surface layer and from the background troposphere generated from the model's dynamical core and initialization. This issue arises from the nature of a global model and is not present when working with a single-column model or LES where the background forcing is prescribed as in Larson et al. (2019). Direct comparisons of findings here to the findings of past studies are thus inherently limited because of this innate difference between the types of models implemented. Future work should test how sensitive the improvements demonstrated in this study are to the surface layer formulation and to the structure of existing background errors that remain unaffected by changes in turbulence parameterization.

In order to improve predictions globally, modelers should identify other regions with strong biases that are thought to result from boundary layer parameterization. Analyses similar to this can prove fruitful for either noting similar errors or determining parameterizations where responses may differ with respect to varying atmospheric regimes. Additional field campaigns reporting detailed observations in these regions alongside LES tailored to those regions would greatly benefit future studies seeking to improve turbulence parameterizations in GCMs.

*Code and data availability.* EUREC⁴A/ATOMIC soundings and derived quantities data used for this project were acquired from https://doi.org/10.25326/137 and are described in Stephan et al. (2020). The version of the Community Atmosphere Model run here was cesm2.2.0 and is available at https://github.com/ESCOMP/CAM. The Cloud Model 1 (CM1) was acquired from George Bryan via https://www2.mmm.ucar.edu/people/bryan/cm1/. ERA5 data used to initialize the hindcasts was downloaded from the Copernicus Climate Data Store (CDS), available at https://www.ecmwf.int/en/forecasts/datasets/reanalysis-datasets/era5. The Betacast software used for hindcast configuration and initialization is available at https://github.com/zarzycki/betacast and is described in Zarzycki and Jablonowski (2015). The version of Betacast used in this manuscript is archived at https://doi.org/10.5281/zenodo.8184863. The data generated for this project (cesm_x*.tar) are available via Penn State's Data Commons at https://doi.org/10.26208/DCSY-HY63. A checkout of the model code (cesm_EUREC4A_sourcetree.tar.gz), case directories for the various configurations (EUREC4A_cases.tar.gz), processed EUREC⁴A/ATOMIC soundings (StephanSoundings.tar), and the scripts (progupwp-GMD-main.zip) used to analyze the data and reproduce the results of this manuscript are available via Zenodo at https://doi.org/10.5281/zenodo.8184357.

## Appendix A: Prognostic momementum derivation and closures

Starting from Eq. 3.3 in Larson (2022):

$$\frac{\partial \overline{u_h' w'}}{\partial t} = \underbrace{-\overline{w}\frac{\partial \overline{u_h' w'}}{\partial z}}_{mean-adv} \underbrace{-\frac{1}{\rho}\frac{\partial \rho \overline{w'^2 u_h'}}{\partial z}}_{turb-adv} \underbrace{-\overline{w'^2}\frac{\partial \overline{u_h}}{\partial z}}_{turb-prod} \underbrace{-\overline{u_h' w'}\frac{\partial \overline{w}}{\partial z}}_{accum} + \underbrace{\frac{g}{\theta_{vs}}\overline{u_h'\theta_v'}}_{buoy-prod} \underbrace{-\frac{1}{\rho}\left(\overline{u_h'\frac{\partial p'}{\partial z}} + \overline{w'\frac{\partial p'}{\partial x_h}}\right)}_{pressure} \underbrace{-\epsilon_{u_h w}}_{dissip} \tag{A1}$$

where $\rho$ is density of air, $g$ is gravity, $\theta_v$ is virtual potential temperature, and $\theta_{vs}$ is a dry base-state potential temperature value. Substituting in Eq. 3.30 from Larson (2022):

$$\underbrace{-\frac{1}{\rho}\left(\overline{u_h'\frac{\partial p'}{\partial z}} + \overline{w'\frac{\partial p'}{\partial x_h}}\right)}_{pressure} \approx \underbrace{-\frac{C_6}{\tau}\overline{u_h' w'}}_{pr1} + \underbrace{C_7\overline{u_h' w'}\frac{\partial \overline{w}}{\partial z}}_{pr2} \underbrace{-C_7\frac{g}{\theta_{vs}}\overline{u_h'\theta_v'}}_{pr3} + \underbrace{C_{uu,shear}\overline{w'^2}\frac{\partial \overline{u_h}}{\partial z}}_{pr4} \tag{A2}$$

where $C_6$, $C_7$, and $C_{uu,shear}$ are all empirical coefficients. Note that $C_{uu,shear}$ is equivalent to $C_{7upwp}$ from Nardi et al. (2022). Eq. A1 Becomes

$$\frac{\partial \overline{u_h' w'}}{\partial t} = \underbrace{-\overline{w}\frac{\partial \overline{u_h' w'}}{\partial z}}_{mean-adv} \underbrace{-\frac{1}{\rho}\frac{\partial \rho \overline{w'^2 u_h'}}{\partial z}}_{turb-adv} \underbrace{-\overline{w'^2}\frac{\partial \overline{u_h}}{\partial z}}_{turb-prod} \underbrace{-\overline{u_h' w'}\frac{\partial \overline{w}}{\partial z}}_{accum} + \underbrace{\frac{g}{\theta_{vs}}\overline{u_h'\theta_v'}}_{buoy-prod}$$
$$\underbrace{-\frac{C_6}{\tau}\overline{u_h' w'}}_{pr1} + \underbrace{C_7\overline{u_h' w'}\frac{\partial \overline{w}}{\partial z}}_{pr2} \underbrace{-C_7\frac{g}{\theta_{vs}}\overline{u_h'\theta_v'}}_{pr3} + \underbrace{C_{uu,shear}\overline{w'^2}\frac{\partial \overline{u_h}}{\partial z}}_{pr4} \underbrace{-\epsilon_{u_h w}}_{dissip} \tag{A3}$$

Rearranging terms with common expressions:

$$\underbrace{\frac{\partial \overline{u_h' w'}}{\partial t}}_{} = \underbrace{-\overline{w}\frac{\partial \overline{u_h' w'}}{\partial z}}_{mean-adv} \underbrace{- \frac{1}{\rho}\frac{\partial \rho \overline{w'^2 u_h'}}{\partial z}}_{turb-adv} \underbrace{- \overline{w'^2}\frac{\partial \overline{u_h}}{\partial z}}_{turb-prod} + \underbrace{C_{uu,shear}\overline{w'^2}\frac{\partial \overline{u_h}}{\partial z}}_{pr4} \underbrace{- \overline{u_h' w'}\frac{\partial \overline{w}}{\partial z}}_{accum}$$

660

$$\underbrace{+ C_7 \overline{u_h' w'}\frac{\partial \overline{w}}{\partial z}}_{pr2} + \underbrace{\frac{g}{\theta_{vs}}\overline{u_h' \theta_v'}}_{buoy-prod} \underbrace{- C_7 \frac{g}{\theta_{vs}}\overline{u_h' \theta_v'}}_{pr3} \underbrace{- \frac{C_6}{\tau}\overline{u_h' w'}}_{pr1} \underbrace{- \epsilon_{u_h w}}_{dissip} \quad (A4)$$

Combining like terms gives us:

$$\underbrace{\frac{\partial \overline{u_h' w'}}{\partial t}}_{} = \underbrace{-\overline{w}\frac{\partial \overline{u_h' w'}}{\partial z}}_{mean-adv} \underbrace{- \frac{1}{\rho}\frac{\partial \rho \overline{w'^2 u_h'}}{\partial z}}_{turb-adv} \underbrace{- (1-C_{uu,shear})\overline{w'^2}\frac{\partial \overline{u_h}}{\partial z}}_{turb-prod} \underbrace{- (1-C_7)\overline{u_h' w'}\frac{\partial \overline{w}}{\partial z}}_{accum}$$

$$\underbrace{+ (1-C_7)\frac{g}{\theta_{vs}}\overline{u_h' \theta_v'}}_{buoy-prod} \underbrace{- \frac{C_6}{\tau}\overline{u_h' w'}}_{pr1} \underbrace{- \epsilon_{u_h w}}_{dissip} \quad (A5)$$

Further, the closure used for $\rho \overline{w'^2 u_h'}$ is closed with Eq. 5 in Larson et al. (2019):

665
$$\frac{\partial}{\partial z}(\rho \overline{w'^2 u_h'}) \approx \frac{\partial}{\partial z}\left(a_1 \frac{\overline{w'^3}}{\overline{w'^2}}\overline{u_h' w'}\right) \quad (A6)$$

the closure for $\overline{u_h' \theta_v'}$ is as in Eq. 33 in Golaz et al. (2002):

$$\overline{u_h' \theta_v'} = \overline{u_h' \theta_l'} + \frac{1-\epsilon_0}{\epsilon_0}\theta_0 \overline{u_h' q_t'} + \left(\frac{L_v}{c_p}\left(\frac{p_0}{p}\right)^{\frac{R_d}{c_p}} - \frac{1}{\epsilon_0}\theta_0\right)\overline{u_h' q_l'} \quad (A7)$$

where $\theta_l$ is the liquid water potential temperature, $R_d$ and $R_v$ are the gas constants for dry air and water respectively, $\epsilon_0 = \frac{R_d}{R_v}$, $C_p$ is the heat capacity of dry air at constant pressure, $L_v$ is the latent heat of vaporization of water, $P_0$ is a reference pressure, 670 $\theta_0$ is a reference potential temperature, $r_l$ is liquid specific water content, and $r_t$ is total specific water content. $\overline{u_h' \theta_l'}$ and $\overline{u_h' q_l'}$ are in turn closed with Eq. 9 in Larson et al. (2019):

$$\overline{u_h' \psi'} = \left(\frac{\tau}{C_{u\psi}^{pi}}\right)\left(-\overline{u_h' w'}\frac{\partial \overline{\psi}}{\partial z} - (1-C_{u\psi}^{ps})\overline{w' \psi'}\frac{\partial \overline{u_h}}{\partial z}\right) \quad (A8)$$

The variable $\psi$ here represents either $\theta_l$ or $r_t$ and the constants $C_{u\psi}^{pi}$ and $C_{u\psi}^{ps}$ are set equal to $C_6$ (2) and 0 respectively. Finally, the closure used for $\epsilon_{u_h w}$ is setting it to 0 as in Eq. 3.31 in Larson (2022):

675
$$\underbrace{- \epsilon_{u_h w}}_{dissip} \approx 0 \quad (A9)$$

*Author contributions.* SG: literature review, data organization and cleaning, data analysis, figure generation, results interpretation, writing. CZ: conceptualization, simulation generation, proofreading and formatting, project administration, supervision, and funding acquisition.

*Competing interests.* The contact author has declared that none of the authors has any competing interests.

*Acknowledgements.* SG and CZ would like to thank Ying Pan, Jerry Y. Harrington, Kyle Nardi, and Vince Larson who provided comments
on a draft of this work. They also greatly appreciate the comments from three anonomyous reviewers that improved the quality of the
manuscript. SG would also like to thank Allen Mewhinney and Marley Majetic for their suggestions, encouragement, and help with coding.

This research is jointly funded as part of a Climate Process Team (CPT) under Grant AGS-1916689 from the National Science Founda-
tion (NSF) and Grant NA19OAR4310363 from the National Oceanic and Atmospheric Administration (NOAA). The authors acknowledge
computing support from Cheyenne (doi:10.5065/D6RX99HX) provided by NCAR's Computational and Information Systems Laboratory,
sponsored by the NSF. Additional data analysis for this research was performed on the Pennsylvania State University's Institute for Compu-
tational and Data Sciences' Roar supercomputer.

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
