# Peer review of "Using EUREC4A/ATOMIC Field Campaign Data to Improve Trade-Wind Regimes in the Community Atmosphere Model"

_EGUsphere, 2023_

## Referee Comment (RC3)

**Using EUREC4A/ATOMIC field campaign data to improve trade-wind regimes in the Community Atmosphere Model**
**Graap & Zarzycki** (2023)

The study seeks to assess improvements to the prediction of tropical shallow cumulus regimes by modifying CLUBB to allow for counter-gradient momentum fluxes (/prognostic momentum fluxes). Overall, it is important work and I found the paper well written and fairly straightforward to follow. I have a few comments that require revisions before the article should be published, but I consider them minor.

- Lines 30-33: "Changes in low cloud fractions… The Hadley cell" – it's unclear how the Hadley cell links to the opening/topic sentence of this paragraph. Suggest adding language to more cleanly transition between the two sentences.
- Lines 149-152: Was the CMIP6 version using 58L CAM, with the refined resolution in the BL, and with the SE grid? My impression is that this might be different from the original CESM2 release version.
- Lines 167-168: "C6 and C7 are also tunable constants, although they are left as 4 and 0.5, respectively, for all simulations here." – are these the default values of C6 and C7 that CLUBB uses out of the box?
- Lines 236-238: "At each of these 10-meter levels, state variable values meant to represent model output are calculated…" – the phrasing here is a bit confusing. The state variables *are* the model output, no? Is the point ultimately that model output is interpolated as a linear vertical distance-weighted average for every 10-meter observation?
- Line 284: "This along with observations in our study being qualitatively similar to the LES-derived profiles in L19…" – Is this implying that the limited observations of u'w' are in line with the LES profiles of L19? I'm confused by the use of "observations" here, which seems contrary to what was stated in the paragraph before.
- Lines 290-291: "In x101, v'w' is also about half as negative at altitudes between 300 m and 2 km." Would be helpful to note that this refers to Figure 1d, not 1c.
- Lines 302-310: I'm not sure what the discussion of the Ekman spiral in the atmosphere lends to this study in particular. Perhaps draw draw a clearer link or consider removing most of this?
- Figure 2: It seems that panels (a) and (b) are just repetition of Figure 1 (a) and (c); is there a way to combine them then to reduce redundancy? Would also be good to name in the caption which panels refer to which part (i.e., "Vertical profiles of means (a-c),…") even though the axis labels are fairly clear.
- Lines 314-315: "It can be seen that although x101 has a stronger jet maximum than x001, it has a reduced maximum easterly bias when compared to x001 since its jet placement matches observations better." – this feels redundant as well, since the smaller bias maximum was noted when discussing Fig 1. Combining Figs 1 & 2 might make this a bit easier to discuss with less repetition.

- Line 317: "near the jet maximum" – is this near the observed jet max, or the model simulated?
- Lines 317-318: "The remainder of the RMSE profiles are quite similar…" – they're nearly identical for $v$, but for $u$ it looks like the simulations are fairly different throughout the vertical; maybe a more nuanced statement is warranted?
- Figure 3: "The vertical axis is a rough estimate of the pressure level of the model output" – could you be more specific? Is this the hybrid coordinate pressure?
- Lines 335-336: "Most points with negative $K_{eff}$ in x101 are above this threshold…" – how far above the threshold do these points typically lie? Is there a large spread in the value, and values are often much larger than the threshold, or are values often close to the value (and perhaps thus the findings are sensitive to the choice of cutoff)?
- Lines 347-348: "Confidence is added to this hypothesis by…" – I see the discussion of LES results with different forcing (i.e., Helfer et al.), but does this refer to other studies that use the EUREC4A/ATOMIC forcing? Would be good to discuss/cite those if so.
- Figure 4 and related discussion: Are these differences in theta and Q profiles statistically significant?
- Line 382: A better qualitative match, yes; is this not a better quantitative match as well?
- Lines 382-385: Suggest adding an in-text reference to panels of Fig 6 as they're discussed. In terms of reducing theta/Q biases in the x200 runs, isn't this to be expected when any run is tuned to better match the ERA5 results? It seems that potential bias reductions in these runs could be driven more by the tuning than by the formulation of L/prognostic momentum.
- Figure 7: It looks like the experimental L formulation has a substantial impact on $u$ biases in the lowest 1 km. In 001 and 101, negative biases extend to the surface, but those seem to be removed with the L cases. Is there a reason for that? Is the L formulation most sensitive closest to the surface?
- Line 404: "Tropospheric" should probably be "troposphere".
- Line 410: "between 200 and 2 km" – should read 200 m and 2 km.
- Figure 8: Missing legend
- Lines 424-425: "do demonstrate a likely connection between the prediction of upgradient fluxes and modifications to various terms in the vertical momentum flux budget" – Could you elaborate on this a bit more for clarity? How does this tie into the vertical momentum flux budget terms? It seems that this is just the prediction of upgradient occurrence in the figure.
- Fig 10: Would be helpful to have additional percentages labeled, not just 100% and "same" (and perhaps same should be written as 0%?). Overall the colorbar combined with the actual bias values in the boxes is a little confusing. It would seem for example that the bias in x101 for Mixing Ratio should be not quite the darkest red (it's not a doubling of the bias), but It's the same color as x201, which is more than a doubling of the bias…
- Lines 445-446: It's worth noting that although "the greatest improvements are seen in $u$ and $U\_h$, there's a stronger degradation in $Q$ when you add in the experimental L

calculation. Would be a more balanced description of the results, at least; elaboration would be great.

- Fig 11: Please add additional colorbar markers, as for Fig 10.
- Lines 489-490: "One of the most notable…" – suggest adding a parenthetical reference to guide the reader exactly where to see this. So maybe at the end, add "(solid brown line in Fig. 12)"? Would help in additional sentences of this paragraph as well.
- Lines 495-496: "…could be leading to changes in atmospheric stability…" – any evidence that could be added to support this?
- Lines 528-536: "This study is a targeted regional investigation and as such, the improvements seen here cannot necessarily be generalized to the global climate system without further exploration…" – This is a really important caveat, and I appreciate the discussion surrounding it. The question arises then – why not use these simulations to evaluate global performance? You have the full global output, so could this dataset be a tool for exploring additional regions/field campaigns, and more generally for looking at global biases? It may be beyond the scope of this particular study, but is it something that's targeted for future work or are the runs not suitable for that analysis?

---

## Author Comment (AC1)

Reviewer #1

This paper performs hindcast CAM simulations for data taken during the EURECA field campaign to attempt to improve the representation of the boundary layer in trade-wind regimes by testing experimental versions of the CLUBB parameterization that include upgradient fluxes for momentum. Overall, I found this to be an interesting paper that addresses an issue that typically receives little attention. Additionally, I found this manuscript to be exceptionally well written, which always makes the job of a reviewer much easier and should not go unnoticed. I do feel this is worthy of publication, after the authors consider some of the following points which I feel would improve the presentation of results and perhaps increase the impact.

Thank you for offering your time to review this manuscript and constructive feedback. Please see point-by-point responses below.

Unlike the Larson et al. (2019) paper, the authors here find that the boundary layer thermodynamic structure is improved with their experimental configurations. I feel it is important to show what impact this has on the simulation of clouds. If the impact is negligible then that could be displayed in a figure or two and would be a worthy reference point for other studies. If the change is more significant then perhaps this should be highlighted a bit more (but I understand this was never the intended focus of the paper) with a discussion of potential implications for climate length simulations and other modeling centers wishing to improve momentum transport.

We have added a figure (Fig. 11) that shows profiles of both the vertical cloud fraction and cloud liquid water content towards the end of the discussion of the experimental length scale. We also add a brief discussion in the text which is reproduced below. However, as the reviewer notes, we emphasize that this isn't a core motivation for this project (and likely merits significant further exploration) so we show it as descriptive and leave a more detailed interpretation for future work.

The following text has been added to the manuscript (in addition to the aforementioned Fig. 11).

*"While the specific focus of this work is on the transport of momentum, we show vertical profiles of cloud liquid and cloud fraction in Fig. 11 since a key motivation for understanding boundary layer processes in this region is to improve the representation of low clouds in Earth system models (and their associated forcing on the climate system). When prognostic momentum is turned on (ED-O to PM-O) both cloud liquid and cloud fraction decrease. A decrease in the height of peak cloudiness also occurs. Both of these changes tend to represent a better agreement with the CM1 LES results, although we stress that we have not undertaken a rigorous comparison with observations from a cloud perspective. Nonetheless, we do note these results are qualitatively similar to those published in Narenpitak et al. (2021) and Schulz and Stevens (2023). Turning on the experimental length scale formulation (ED-X and PM-X) results in an increase in cloud liquid and a further reduction in the height of the peak cloudy layer. Both of these further improve the correspondence of the profile shape to the CM1 results, although both liquid and fraction are overestimated in magnitude relative to the LES run. Somewhat interestingly, going from ED-X to PM-X increases cloud liquid, which is counter to the same*

*change using the original length scale formulation (ED-O to PM-O). While this is just a cursory look at cloud fields, it would imply that changes in the treatment of momentum fluxes also feed back into cloud fields, but that the updated treatment of τ may play an equally or larger role. This is unsurprising given that τ appears in many equations throughout CLUBB, not just those associated with momentum (Golaz et al., 2002). These cloud responses to both momentum treatment and length scale formulation are complex and merit additional evaluation and calibration."*

I am not a fan of the naming convention for the simulations (x001, x101, etc,). Why not refer to "x001" simply as "CNTL" and devise similarly brief but easily distinguishable names for the other sets of simulations?

We agree that the original naming convention wasn't overly clear – it was our internal organization strategy, and we should have essentially built a lookup table in our plotting scripts for these in the original submitted manuscript. We have renamed the experiments based on this suggestion. x001 is now "ED-O" (eddy diffusivity, original length scale), x101 is now "PM-O" (prognostic momentum, original length scale), x204 is now "PM-X" (prognostic momentum, experimental length scale), and x304 is now "ED-X" (eddy diffusivity, experimental length scale). This is described in the text and we have also added a table (Table 1) to help readers interpret the experiment IDs.

The authors present results for control CAM, CAM with prognostic momentum fluxes, and CAM with prognostic momentum fluxes but revised length scale definition. I feel it is also very important to present results from CAM with only the revised length scale modification (essentially the Guo et al. 2021 configuration). Having this data point is essential to help the reader tease out the relative contribution of improvement stemming from this modification alone.

In the revised manuscript, we have added another configuration (ED-X) that does this. We do not apply the updated prognostic momentum flux formulation but do update the length scale formulation. This now gives us four simulations (see below response re: the other sensitivity runs) spanning the full space of T/F for both the prognostic momentum and experimental length scale combinations.

I am confused why the authors chose to perform tuning experiments for the configurations that include the revised length scale definition. While I understand it is important to exploit model sensitivity to tunable parameters, it is odd that they chose to do it for this configuration but not experiment x101 (which also introduces new tunable parameters) and this makes the article feel a bit disjointed. In addition, presenting results for all four x20 simulations is a bit redundant and cumbersome. If the authors feel it pertinent to include these tuning simulations perhaps they could just show the average result (and summarize the sensitivity/spread in a short section with a figure or two)? In this instance I feel the authors would also need to add a strong justification why they felt it necessary to do a tuning suite for this particular configuration while neglecting to do so for the others (or perhaps add a tuning suite for the other configurations). Overall, though (unless justification is provided) the main purpose of this paper didn't seem to be about finding optimal tuning parameters nor exploiting CLUBB's sensitivity to them. Therefore, I'm inclined to suggest the authors just present results from one realization of x201 (with whatever the default

parameters are from Guo et al. 2021) and briefly summarize that the sensitivity to tunable parameters was explored and highlight the most important points of that analysis.

In response to this (and the above comment) we have decided that having four versions of the experimental length scale indeed does muddy the picture a bit. Based on this feedback (and the above addition of ED-X), we choose to only keep one set of parameters for the experimental length scale (formerly x204, now PM-X). We feel that this tightens the manuscript and allows for focus on the two experimental changes explored here.

---

## Author Comment (AC2)

Reviewer #2

By conducting a series of CAM hindcast simulations during the EUREC4A/ATOMIC field campaign in the Tropical Atlantic in early 2020, this study points out that the prognostic treatment of momentum flux improves the simulation of tropical trade-wind to a greater extent, compared with the default CAM-CLUBB. This study improves the CLUBB scheme by further implementing a generalized calculation of the turbulent length scale, which reduces model bias and RMSE relative to observation. Recently, more and more studies have noticed that momentum fluxes can be upgradient in shallow convection regimes, and it is challenging to parameterize them. In my opinion, this study is interesting. This paper also gives us some insights on how to further improve the prognostic treatment of momentum flux and simulation of trade-wind regimes in global models.

In addition, this paper is well-written. I therefore recommend the publication of this paper in GMD after some revisions.

Thank you for your review and thoughtful comments.

I noticed that Guo et al. (2021) applied additional damping of non-cloudy layers stable stratification to scaler flux and also w'3 in the stable layers, because of gravity-wave dispersion under cloudy conditions (please refer to eq23 and 24 in Guo et al. 2021). This might be of some help in the simulation of shallow convections and stratocumulus. The case of momentum flux may be similar to the scaler flux, should an additional clear sky damping be considered in Eq 6 as well? And, will it further improve the results?

This is a good thought and certainly something we'd like to explore in the future. For this exploration, we aimed to compare the potential changes in CLUBB with respect to the EUREC4A/ATOMIC field campaign observations, and therefore the exploration only included a potential subset of modifications to CLUBB (including the simple "taus" formulation in Guo et al., 2021). We are working with Vince Larson and other researchers at the University of Wisconsin-Milwaukee and the National Center for Atmospheric Research to explore the utility of additional damping mechanisms.

We have added a sentence to the discussion: "*While we only evaluated a subset of targeted dissipation permitted by this experimental length scale treatment, other possibilities, such as the additional damping of the third-order moment of vertical velocity in stable layers described in Guo et al. (2021), merit further study.*"

C6 is used in scale flux and momentum flux, which are very important for the return-to-isotropy terms. But, I also noted that C6=C6b= 1 in Guo et al. (2021). So, the Taus could have full control over the return-to-isotropy in the momentum flux equation. This study uses the default CAM setting of C6=4 (and I assume C6b=6), which would involve the skewness function in the calculation of the pressure term (Larson 2022), which overlaps with the role of the Tau scheme. It creates some difficulties in understanding the performance and improvement of parameterization. I suggest that the authors set C6=C6b=1 for the experiments to simplify the problem.

Thanks for raising this point. We actually set C6 = 2 as constant (i.e., no skewness function) in the experimental length scale runs. This is based on personal communication with CLUBB's developer Vince Larson. This has been noted in the text and the correspondence from Vince Larson is reproduced below for posterity. We have also updated the text to properly define C6 for the "O" (original length scale) runs (i.e., that we retain the CAM6 default settings).

"The value of 2 is a historical artifact, and it is preserved in our code so that we can recover [the Guo et al., 2021] tuning.

Someday we'll get rid of it, but the key point is not the particular value of C6, but that if you're tuning, you should tune the C_invrs_... parameters and not C6, C1, etc., because the latter parameters are redundant, and we don't want the space of tuning parameters to be too large."

The difference between the X101 and X204 horizontal momentum fluxes budgets is remarkable. The budgets of horizontal momentum fluxes in X204 qualitatively resemble those of BOMEX and RICO in LES (e.g. Figure 7 and 9 in Larson et al., 2019), with the buoyant term predominantly balancing the turbulence production term. I'm curious what causes this big change, is it due to the tuning listed in Table 1 or the model structure changes? This study does give us some explanations, but it would be better to add some more discussions and show more turbulent profiles that help the reader to better understand the improvement, such as $w^2$ , $u^2$ , $v^2$ and scaler fluxes. $u^2$ , $v^2$ would also benefit directly from the prognostic treatment of momentum flux.

We agree! Unfortunately, performing a much deeper dive than what we have shown here would be difficult without re-running many of the simulations with additional diagnostics. However, we do more closely link the results of Larson et al., 2019 and do also highlight that we should more closely look at these budget terms in future work to better understand why the set of tunings in PM-X appears to better produce LES profiles for test cases with similar dynamical setups to the EUREC4A/ATOMIC atmospheric environment.

The text in this section has been modified to read: *"Another notable difference is the changing of the sign of the buoyancy production term (solid blue lines) from weakly negative in PM-O to notable positive below 700 m, and negative above in PM-X, particularly in the zonal momentum budget. This is also qualitatively consistent with the BOMEX LES budgets in L19 (their Fig. 7) which lends credence to process-level improvement in the PM-X runs. We hypothesize that this may be related to increased stratification in the θ profile in PM-X making vertical transport or air parcels due to buoyancy more difficult in the lowest 700 m. In that case, improvement in the thermodynamic profile in PM-X could be leading to changes in atmospheric stability (e.g., note the differences in the change in θ with height in Fig. 6a), which in turn lead to changes in buoyant production of uh'w' which then feeds back to changes in the dynamic profiles. Since downgradient diffusion corresponds to a simple balance between turbulent production and return-to-isotropy, the fact that the buoyancy term is so large in PM-X could explain the enhanced upgradient fluxes in Fig. 9. We admit this is speculative, however, and experiments with more constrained model configurations (e.g., single column, nudged runs) and voluminous diagnostics would be helpful in providing deeper insight, including more detailed consideration*

*of other turbulent quantities such as the vertical fluxes of temperature and moisture as well as variances (e.g., u'2 and v'2 would be directly affected by the additional of prognostic momentum to CLUBB)."*

Also, how were the parameters in x201-x204 determined (Table 1)?

These were defined using a Nelder-Mead optimization strategy described below. As suggested by Reviewer #1 we now only include one set of tuning parameters to simplify the paper and allow for a cleaner evaluation of the prognostic momentum and experimental length scale formulations. The original Table 1 has been removed and replaced with in-text notation of the tuning coefficients. The text now reads:

*"We determine tuning coefficients for this configuration using a Nelder-Mead optimization (Nelder and Mead, 1965). Specifically, a set of very short (48-hour) hindcasts initialized on January 1st, 2012 is run, optimizing various tunable parameters in CLUBB to minimize the difference in the predicted wind field after 2 days when compared against ERA5 reanalysis at the same time. Optimization is completed relative to global ERA5 reanalysis data rather than the local EUREC4A/ATOMIC data to ensure a reasonable global simulation."*

The names of the experiments in the main text make it a bit difficult for me to read, could you please give them more visual names?

The simulations have been renamed to include the prefix "ED" if they diagnose momentum fluxes via eddy diffusivity or "PM" if they apply the prognostic momentum formulation described in Section 2. We also include a suffix of either "O" if the model uses the original length scale formulation published by Golaz et al., 2002, or "X" if using the experimental "taus" formulation from Guo et al., 2021. See our new Table 1 and our response to Reviewer #1 regarding the naming convention, as well.

Figure 8, legend explanation?

We have added a legend to the center panel of Fig. 8 that defines the specific lines in these plots.

If possible, it would be better to add some LES results to the plots, like in Figures 1, 5, and 12?

We have set up a configuration of Cloud Model 1 (CM1) developed by George Bryan (NCAR) to run a "EUREC4A/ATOMIC" LES case. This LES case is built off the traditional BOMEX test case published in Siebesma et al., (2003, J. Atm. Sci.) but with updates to more accurately handle the large-scale environment observed during the EUREC4A/ATOMIC field campaign. Qualitatively these results agree with LES simulations published using the BOMEX test case, but the state field profiles we simulate here are better matched to the actual radiosonde observations. We describe the updates in the manuscript so that other researchers can reproduce these results and the domain-averaged model data (along with CM1 namelists and configuration files) are included in the project's GitHub repository archived at Zenodo. Like the CAM simulations, the LES simulation also shows a region of countergradient fluxes, as has previously been demonstrated in recent BOMEX LES simulations (e.g., Larson et al., 2019, Dixit et al., 2020).

Line 410, page 18, "between 200 and 2 km", do you mean 200 meters?

Yes, you are correct -- we accidentally omitted the "m" label here. This is fixed.

---

## Author Comment (AC3)

Reviewer #3

The study seeks to assess improvements to the prediction of tropical shallow cumulus regimes by modifying CLUBB to allow for counter-gradient momentum fluxes. Overall, it is important work and I found the paper well written and straightforward to follow. I have a few comments that require revisions before the article should be published, but I consider them minor. I have attached specific comments in a PDF.

*Thank you for your positive comments and subsequent feedback. Please see responses below.*

Lines 30-33: "Changes in low cloud fractions… The Hadley cell" – it's unclear how the Hadley cell links to the opening/topic sentence of this paragraph. Suggest adding language to more cleanly transition between the two sentences.

*Agree this transition was awkward. This has been reworded.*

Lines 149-152: Was the CMIP6 version using 58L CAM, with the refined resolution in the BL, and with the SE grid? My impression is that this might be different from the original CESM2 release version.

*Yes, you are correct, we use a version that differs slightly from the official out-of-the-box CAM6 release -- mainly that we use the spectral element dynamical core instead of the finite-volume and 58 vertical levels instead of 32. To clarify this for the readers we have changed this to read:*

*"The version of CAM studied here is CAM version 6 (Bogenschutz et al., 2018; Gettelman et al., 2019). This corresponds to the configuration of CAM in the CESM version 2 release (Danabasoglu et al., 2020) that was used to generate the simulation submitted to the Coupled Model Intercomparison Project version 6 (CMIP6), with two differences. First, we use the spectral element (SE) dynamical core (Lauritzen et al., 2018) on an unstructured cubed-sphere grid with nominal 1◦ (111km, also referred to as CAM-SE's ne30np4 grid) horizontal grid spacing. This is in lieu of the CAM6 default finite-volume dynamical core. Second, we use 58 vertical levels with finer grid spacing in the atmospheric boundary layer compared to CAM6's default 32 layers."*

Lines 167-168: "C6 and C7 are also tunable constants, although they are left as 4 and 0.5, respectively, for all simulations here." – are these the default values of C6 and C7 that CLUBB uses out of the box?

*Yes, we have modified the text to better clarify this, but to answer here, we preserve the default CLUBB settings in the CAM6 release in the "O" runs.*

Lines 236-238: "At each of these 10-meter levels, state variable values meant to represent model output are calculated…" – the phrasing here is a bit confusing. The state variables are the model output, no? Is the point ultimately that model output is interpolated as a linear vertical distance-weighted average for every 10-meter observation?

To clarify this we have replaced "… [at] each of these 10-meter levels, state variables meant to represent model output are calculated by taking the linear vertical distance-weighted average of those values reported at the nearest two model levels." with "… state variables from model output are linearly interpolated to each of these 10-meter levels by taking the linear vertical distance-weighted average of those values reported at the nearest two model levels."

Line 284: "This along with observations in our study being qualitatively similar to the LES-derived profiles in L19…" – Is this implying that the limited observations of u'w' are in line with the LES profiles of L19? I'm confused by the use of "observations" here, which seems contrary to what was stated in the paragraph before.

"Observations" in line 284 refers to observations of u and v, not  like the bulk of the paragraph. We acknowledge this was unclear and have updated the sentence to read: "*This along with observations of u and v in our study being qualitatively similar to those in the LES-derived profiles in L19…*"

Lines 290-291: "In x101, v'w' is also about half as negative at altitudes between 300 m and 2 km." Would be helpful to note that this refers to Figure 1d, not 1c.

We have added a figure reference to this sentence to refer to Fig. 1d.

Lines 302-310: I'm not sure what the discussion of the Ekman spiral in the atmosphere lends to this study in particular. Perhaps draw a clearer link or consider removing most of this?

Removed.

Figure 2: It seems that panels (a) and (b) are just repetitions of Figure 1 (a) and (c); is there a way to combine them then to reduce redundancy? Would also be good to name in the caption which panels refer to which part (i.e., "Vertical profiles of means (a-c),…") even though the axis labels are fairly clear.

The reviewer is correct that those two panels are reproduced. We cannot come up with a clean way of eliminating them from Fig. 2, however, we add text that notes they are identical to part of Fig. 1. We also like the idea of adding parentheticals regarding the panel labels and have amended the caption to include the suggestion.

Lines 314-315: "It can be seen that although x101 has a stronger jet maximum than x001, it has a reduced maximum easterly bias when compared to x001 since its jet placement matches observations better." – this feels redundant as well, since the smaller bias maximum was noted when discussing Fig 1. Combining Figs 1 & 2 might make this a bit easier to discuss with less repetition.

The text has been amended to reduce the redundancy. We also specifically call out individual panels in Fig. 2 when discussing the bias and RMSE results to focus the reader's attention as they make their way through the paragraph.

Line 317: "near the jet maximum" – is this near the observed jet max, or the model simulated?

After reviewing the figure, "*near the jet maximum*" has been replaced by "*in the region immediately above the modeled jet maximum (roughly 1 to 2 km altitude).*"

Lines 317-318: "The remainder of the RMSE profiles are quite similar…" – they're nearly identical for v, but for u it looks like the similarities are fairly different throughout the vertical; maybe a more nuanced statement is warranted?

True, "*[the] remainder of the RMSE profiles are quite similar,*" has been changed to "*[both] the RMSE profile for v and the RMSE profile for u far from the modeled/observed jet maxima are quite similar.*"

Figure 3: "The vertical axis is a rough estimate of the pressure level of the model output" – could you be more specific? Is this the hybrid coordinate pressure?

To better specify, we have added a note "*… levels here are taken from a column at a single time, making the pressure levels estimates, since the hybrid pressure coordinates change depending on elevation and surface pressure. In this situation, this is a reasonable estimate since all balloons were launched from near sea level and almost all drifted over the open ocean in fair weather conditions.*"

Lines 335-336: "Most points with negative Keff in x101 are above this threshold…" – how far above the threshold do these points typically lie? Is there a large spread in the value, and values are often much larger than the threshold, or are values often close to the value (and perhaps thus the findings are sensitive to the choice of cutoff)?

We have included a histogram of $K_{eff}$ values in Fig. 10. The number of points lying below this threshold is 0.1-0.2%. When this filter is included, the mean (median) wind shear for negative $K_{eff}$ points is 0.52 (0.43) m/s per km and 2.2 (1.6) m/s per km for PM-O and PM-X respectively, indicating the majority of the negative $K_{eff}$ points are generated in the presence of wind shears at least double this threshold.

Lines 347-348: "Confidence is added to this hypothesis by…" – I see the discussion of LES results with different forcing (i.e., Helfer et al.), but does this refer to other studies that use the EUREC4A/ATOMIC forcing? Would be good to discuss/cite those if so.

We have added some additional references to recent work using LES with ATOMIC/EUREC4A data, albeit primarily focused on shallow convective cloud structures.

"*We also refer interested readers to Narenpitak et al. (2021), Dauhut et al. (2023), and Schulz and Stevens (2023), all of which performed LES simulations using a variety of configurations to investigate the distributions and organization of shallow convective clouds during the EUREC4A/ATOMIC study period.*"

Figure 4 and related discussion: Are these differences in theta and Q profiles statistically significant?

Yes, even though the differences are quite small, they are systematic across a vast number of soundings. To test this, we performed a paired t-test at each altitude in the sounding across an ordered list of the soundings in the observational comparison. The majority (>50%) of altitudes were different at alpha = 0.05 level (some altitudes did not register as statistically significant such as θ just below 1 km, where the profiles lie nearly on top of one another.

To highlight this in the text we add: "*These differences in thermodynamic profiles are not as large as the differences in the momentum profiles but do exist. In fact, these differences are still significant at most altitudes when performing a paired Student's t-test across the model profiles included in Fig. 4 (92% (72%) of altitude bins in the θ (Q) profiles significantly differ between ED-O and PM-O at the α = 0.05 level).*"

Line 382: A better qualitative match, yes; is this not a better quantitative match as well?

Yes, the experimental length scale runs match observations of v better at most altitudes both in terms of absolute error and in structure, the word "qualitative" is removed from this sentence.

Lines 382-385: Suggest adding an in-text reference to panels of Fig 6 as they're discussed. In terms of reducing theta/Q biases in the x200 runs, isn't this to be expected when any run is tuned to better match the ERA5 results? It seems that potential bias reductions in these runs could be driven more by the tuning than by the formulation of L/prognostic momentum.

Figure panel references have been added in this section. Although the model configurations were tuned to match ERA5 state variables, we only attempted to match *u* and *v* globally. It was not known ahead of time if small variations in these parameters would noticeably affect model predictions both for thermodynamic quantities and quantities in the study region.

Figure 7: It looks like the experimental L formulation has a substantial impact on u biases in the lowest 1 km. In 001 and 101, negative biases extend to the surface, but those seem to be removed with the L cases. Is there a reason for that? Is the L formulation most sensitive closest to the surface?

The bias reduction in u in the experimental L cases can be seen in Fig. 5 as a rightward shift of the mean wind profile in the experimental L runs relative to the eddy diffusivity (ED-O) and original length scale prognostic runs (PM-O). This bias reduction is most notable relative to PM-O below 1.2 km where PM-O overestimates the strength of the jet maximum, and relative to ED-O above 1.2 km where ED-O diffuses momentum too high. The bias reduction appears to be strongest in the lowest 1 km in Fig. 7 likely because this region has lower bias to begin with and thus the same linear reduction is a larger percent reduction, and because the color bar begins to change from red to blue making the reduction more noticeable.

Line 404: "Tropospheric" should probably be "troposphere".

Correct, this has been updated.

Line 410: "between 200 and 2 km" – should read 200 m and 2 km.

Thank you for catching this. Corrected.

Figure 8: Missing legend

Fixed.

Lines 424-425: "do demonstrate a likely connection between the prediction of upgradient fluxes and modifications to various terms in the vertical momentum flux budget" – Could you elaborate on this a bit more for clarity? How does this tie into the vertical momentum flux budget terms? It seems that this is just the prediction of upgradient occurrence in the figure.

We agree this was unclear. It has been reworded to read: "*We emphasize that these more frequent predictions of upgradient fluxes are not necessarily more accurate, however, they do demonstrate a likely connection between the prediction of countergradient fluxes and modifications to the turbulent dissipation in CLUBB. That is, in the `PM' simulations, changes to the turbulent length scale aimed at improving the shape of the near-surface u and v profiles can further enhance the generation of upgradient momentum fluxes.*"

Fig 10: Would be helpful to have additional percentages labeled, not just 100% and "same" (and perhaps same should be written as 0%?). Overall the colorbar combined with the actual bias values in the boxes is a little confusing. It would seem for example that the bias in x101 for Mixing Ratio should be not quite the darkest red (it's not a doubling of the bias), but It's the same color as x201, which is more than a doubling of the bias…

We have relabeled "same" as "no change" in response to this comment – we agree that fits better.

We have also added additional labels to the bar rather than just the minimum, maximum, and central points as before.

Reported values here are rounded to two decimal places, so moving from ED-O to PM-O (from -0.02 to -0.03) is not actually a 50% increase in bias, but indeed a greater than 100% increase in bias (from -0.016 to -0.033). We agree that the coloring is not very informative for the mixing ratio row, but this is described in the text. We have considered a few other options such as standardizing by the mean value, although this artificially scales quantities based on their mean state (e.g., temperature). To preserve the color scale while maintaining consistency across all variables, we choose to leave the color bar as is and further emphasize this fact in the text.

Lines 445-446: It's worth noting that although "the greatest improvements are seen in u and U_h, there's a stronger degradation in Q when you add in the experimental L calculation. Would be a more balanced description of the results, at least; elaboration would be great.

We have added the caveat "*Some bias degradation is seen in these means for v and Q, but these results are not very meaningful as the mean biases for both these variables are small to begin with and the absolute changes in biases between model configurations are small as well.*"

Fig 11: Please add additional colorbar markers, as for Fig 10.

Done.

Lines 489-490: "One of the most notable…" – suggest adding a parenthetical reference to guide the reader exactly where to see this. So maybe at the end, add "(solid brown line in Fig. 12)"? Would help in additional sentences of this paragraph as well.

Thank you for the suggestion. Parentheticals describing the line styles in the figure have been added to this paragraph.

Lines 495-496: "…could be leading to changes in atmospheric stability…" – any evidence that could be added to support this?

We do not have direct evidence, although the thermodynamic profiles in Fig. 6 show changes in $d\theta/dz$ in the different runs (static stability). More constrained models (e.g., single-column or nudged simulations) may help shed light on exactly the mechanics at play, although that is beyond the scope of this study.

We have added a pointer in the text to the θ profiles in this section. We also add "*We admit this is speculative, however, and experiments with more constrained model configurations (e.g., single column, nudged runs) would be helpful in providing deeper insight.*"

Lines 528-536: "This study is a targeted regional investigation and as such, the improvements seen here cannot necessarily be generalized to the global climate system without further exploration…" – This is a really important caveat, and I appreciate the discussion surrounding it. The question arises then – why not use these simulations to evaluate global performance? You have the full global output, so could this dataset be a tool for exploring additional regions/field campaigns, and more generally for looking at global biases? It may be beyond the scope of this particular study, but is it something that's targeted for future work or are the runs not suitable for that analysis

The reviewer is correct in that there is nothing preventing the global results from being evaluated. Some global data is in fact included in the data DOI attached to this manuscript for other researchers who'd like to explore. It's worth noting that some of the specific turbulence quantities used to evaluate the momentum flux budgets are only output over the EUREC4A/ATOMIC study region to reduce the data output burden while the model was being run. However, we do note that we hesitate to undertake a detailed global evaluation given the fact that evaluation and tuning are specifically targeted on the EUREC4A/ATOMIC region and our experience indicates that attempting to optimize tuning parameters for a particular geographic region will almost certainly result in at least some degradation elsewhere in the global simulation (also see Hourdin et al., 2017, BAMS). That said, ultimately the goal of this

work is to have some of the updates described in this manuscript implemented in the global version of CAM used for CMIP-class simulations, so a more detailed understanding of the turbulence budgets in multiple atmospheric environments is an (exciting) target for ongoing and future work.